# RAVEL: Reasoning Agents for Validating and Evaluating LLM Text Synthesis

## Abstract

Large Language Models have evolved from single-round generators into long-horizon agents, capable of complex text synthesis scenarios. However, current evaluation frameworks lack the ability to assess the actual synthesis operations, such as outlining, drafting, and editing. Consequently, they fail to evaluate the actual and detailed capabilities of LLMs. To bridge this gap, we introduce **RAVEL**, an agentic framework that enables the LLM testers to autonomously plan and execute typical synthesis operations, including outlining, drafting, reviewing, and refining. Complementing this framework, we present C3EBENCH, a comprehensive benchmark comprising $1,258$ samples derived from professional human writings. We utilize a "reverse-engineering" pipeline to isolate specific capabilities across four tasks: CLOZE, EDIT, EXPAND, and END-TO-END. Through our analysis of 14 LLMs, we uncover that most LLMs struggle with tasks that demanding contextual understanding under limited or under-specified instructions. By augmenting **RAVEL** with SOTA LLMs as operators, we find that such agentic text synthesis is dominated by the LLM's *reasoning capability* rather than raw generative capacity. Furthermore, we find that a strong reasoner can guide a weaker generator to yield higher-quality results, whereas the inverse does not hold. Our code and data are available at https://anonymous.4open.science/r/ICML2026-RAVEL.

## 1. Introduction

Large Language Models (LLMs) have evolved from simple monolithic auto-regressive generators (Vaswani et al., 2023; Brown et al., 2020; Ouyang et al., 2022) into long-horizon

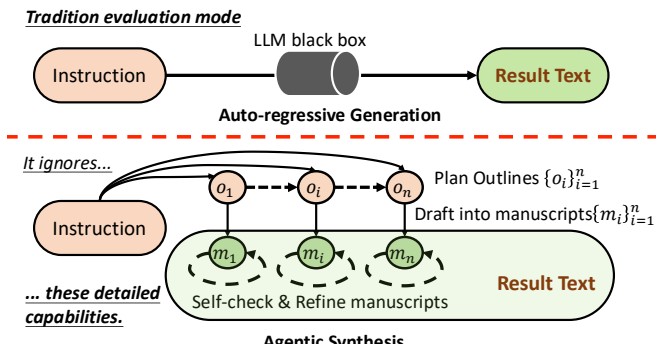

*Figure 1.* An Illustration of the evaluation mode gap.

agents capable of handling complex, sophisticated long-form text synthesis (Yao et al., 2023; Bai et al., 2024; Park et al., 2023; Singh et al., 2025; Gemini Team, Google, 2025; Anthropic, 2025). Most recent frameworks involve LLMs to perform long-form text synthesis in an agentic approach (Yang et al., 2022; Madaan et al., 2023; Schick et al., 2022).

In practical text synthesis scenarios, LLMs have to handle distinct tasks, such as outline planning (Yang et al., 2022; Rashkin et al., 2020), drafting partial sections (Schick et al., 2022; Bai et al., 2024), content-specific editing (Ke et al., 2024; Lin et al., 2024), rather than a simple one-run generation (Wu et al., 2025; Liu et al., 2024). These tasks are usually concurrent and interleaved during the whole synthesis task. However, current evaluation protocols (Zheng et al., 2023; Fu et al., 2023; Kim et al., 2024b; Du et al., 2025) and benchmarks (Wu et al., 2025; Liu et al., 2024; Zheng et al., 2023; Kim et al., 2024a) treat the complex composition as a single run generation, neglecting the intricate capability differences. As a result, they fail to show the actual and detailed capabilities of LLMs during handling real text synthesis.

To align the evaluation framework with the actual LLM capability, we proposed **RAVEL** (**R**easoning **A**gents for **V**alidating and **E**valuating **LL**M text synthesis), a framework that integrates the tested LLM to autonomously plan and execute the synthesis operations by itself, expanding the evaluation scope to more complex scenarios. This agentic setup allows us to evaluate the LLM based on its execution trajectory rather than just the final output. Specifically, we

[1]Anonymous Institution, Anonymous City, Anonymous Region, Anonymous Country. Correspondence to: Anonymous Author <anon.email@domain.com>.

Preliminary work. Under review by the International Conference on Machine Learning (ICML). Do not distribute.

feature four representative synthesis operations (outlining, drafting, reviewing and editing) as the action space $\mathcal{A}$ and formalize the process as a sequential decision process (SDP), where the state $\mathcal{S}$ represents the evolving synthesized textual content. By reasoning and acting (Yao et al., 2023), the LLM evolves the text and terminates the synthesis until the text meets the quality threshold evaluated by itself.

To complement **RAVEL**, we present C3EBENCH (derived from **C**loze, **E**dit, **E**xpand, **E**nd2End), a comprehensive benchmark consisting of $1,258$ samples and expanding four distinct synthesis scenarios: CLOZE, EDIT, EXPAND, END-TO-END. Unlike existing datasets (Wu et al., 2025; Kim et al., 2024a) that pairing text references from elaborated instructions, C3EBENCH employs a "reverse-construction" pipeline to pair professional human references (golden truth) with instructions, in order to recover the complexity and reality of actual text synthesis.

We conduct an extensive empirical analysis of 14 mainstream LLMs, including proprietary models (Singh et al., 2025; Gemini Team, Google, 2025; Anthropic, 2025; Team et al., 2025; GLM-Team, 2025; xAI, 2025; DeepSeek-AI, 2024) and open-source counterparts (Grattafiori et al., 2024; Yang et al., 2025). By leveraging **RAVEL** to roll out execution trajectories of the tested LLMs, we expose critical **behavioral divergences** hidden by static leaderboard rankings. Our results show that most LLMs are capable of handling tasks with detailed and clear specifications from the instructions, yet struggle with scenarios where instructional specifications are limited while contextual understanding is demanded heavily. There is a critical irrelevancy between refinement density and quality gains: while models like Claude-4.5 and Qwen3-32B exhibit high refinement density, they frequently fail to convert iterations into quality gains, contrasting with models (such as Gemini-3 Pro) that make more efficient and effective refinement. Further ablation studies with **RAVEL** demonstrate that synthesis success is dominated by the *LLM's reasoning ability* to plan and critique, rather than the raw generative capacity during refining drafts. In particular, we find that a strong reasoner can effectively guide a weaker generator to success by 39%, whereas a strong generator cannot compensate for the deficient reasoner.

## 2. Preliminaries

### 2.1. Problem Formulation and Notation

To construct realistic, long-horizon synthesis scenarios for evaluating the capabilities of LLMs, we must move beyond static input-output based evaluations. As discussed, text synthesis is inherently iterative and hierarchical. Consequently, we formalize the synthesis task not as single-step generation, but as a Sequential Decision Process (SDP), denoted by the

tuple $\langle \mathcal{Q}, \mathcal{S}, \mathcal{A}, \mathcal{P}, \mathcal{R} \rangle$:

- $\mathcal{Q}$ represents the **query space**. A specific query $q \in \mathcal{Q}$ is defined by a topic $T$ and a style guide $G$.

- $\mathcal{S}$ is the **state space**. A state $s_t \in \mathcal{S}$ at step $t$ is a triplet $s_t = \langle \mathcal{O}, \mathcal{M} \rangle$, where outline $\mathcal{O} = \{o_1, \ldots, o_n\}$ is the set of planned outline nodes, and manuscript $\mathcal{M} = \{m_1, \ldots, m_n\}$ is the synthesized text nodes.

- $\mathcal{A}$ is the **action space** consisting of five operations: $\mathcal{A} = \{\text{outline}, \text{draft}, \text{review}, \text{refine}, \text{finish}\}$.

- $\mathcal{P} : \mathcal{S} \times \mathcal{A} \rightarrow \mathcal{S}$ is the **transition function**, representing the deterministic or stochastic update of the manuscript based on the execution of an action.

- $\mathcal{R} : \mathcal{S} \times \mathcal{Q} \rightarrow \mathbb{R}$ is the **intrinsic reward function**. Crucially, $\mathcal{R}$ is not provided by an external environment oracle; instead, the agent serves as its own *generative oracle* to estimate the quality of $s_t$ relative to $q$.

### 2.2. Optimization Objective

The policy $\pi_\theta(a_t|s_t, q)$ maps the current state and query to an action $a_t$. Because ground-truth feedback is inaccessible during inference for the LLM agent, the process is driven by the internal reward estimator $\mathcal{R}$. Formally, we seek the most efficient trajectory to reach a state $s_T$ that satisfies a quality threshold $\tau$:

$$\min_{a_{0:T}} \quad T \quad \text{s.t.} \quad \mathcal{R}(s_T, q) \geq \tau \tag{1}$$

This formulation enforces a closed-loop refinement process where REVIEW and REFINE actions iteratively improve the manuscript until the quality constraint is met.

## 3. The RAVEL Framework

We instantiate the SDP formulation via **RAVEL**, an environment where the LLM functions as an autonomous agent. Unlike static generation pipelines, **RAVEL** operates on a *Reasoning-Act* paradigm (Yao et al., 2023). At each step $t$, the agent observes the state $s_t$ and samples an action $a_t$ along with execution parameters:

$$(a_t, \text{params}_t) \sim \pi_\theta(s_t, \text{prompt}_{sys}) \tag{2}$$

where $\text{prompt}_{sys}$ encodes the protocol for state transitions and the definition of action primitives. After the agent conducts the action $a_t$, the state $s_t$ is updated to a new state $s_{t+1} \leftarrow \mathcal{P}(s_t, a_t, \text{params}_t)$. Therefore, the holistic trajectory after execution is denoted as $\mathcal{T} = \{(a_t, \text{params}_t), s_{t+1}\}_{t=0}^{T}$. The transition dynamics $\mathcal{P}$ are defined by four specialized action primitives, *Outline, Draft, Review,* and *Refine*, which we detail below.

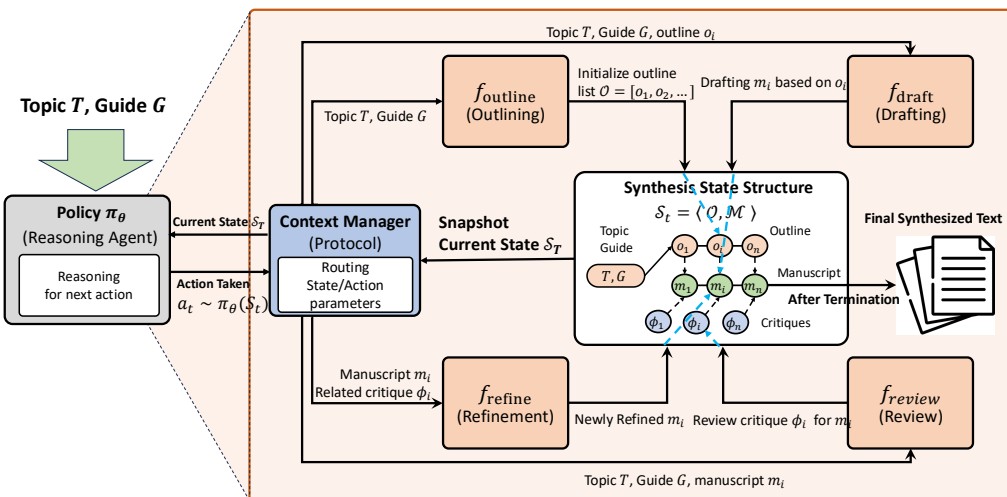

*Figure 2.* **RAVEL**: an evaluation framework that integrates the testing LLM in agentic text synthesis. After receiving the writing instruction, the testing LLM will autonomously reason the synthesis actions to take, and carry out the actions to update its state.

### 3.1. Outlining

The outlining primitive $f_{\text{outline}} : (T, G) \rightarrow \mathcal{O}$ generates the initial structural backbone of the manuscript, where $T$ is the writing topic and $G$ is the genre specification. Each outline node $o_i \in \mathcal{O}$ is defined as a tuple containing textual specifications $t_i$ and a status indicator $\sigma_i$:

$$\mathcal{O} = \{o_i\}_{i=1}^n, \quad \text{where } o_i = \langle t_i, \sigma_i \rangle \qquad (3)$$

The indicator $\sigma_i \in \{\text{pending, drafted, revision\_needed} , \text{completed}\}$ tracks the lifecycle of each node. Upon execution of $f_{\text{outline}}$, all $\sigma_i$ are initialized to "pending". This primitive serves to establish global coherence and structural constraints prior to more specified content synthesis.

### 3.2. Guided Synthesis: Drafting

The drafting primitive $f_{\text{draft}}$ is responsible for generating content for a target outline node. To ensure narrative consistency, we define each manuscript node $m_i \in \mathcal{M}$ as a tuple $\langle t, \phi \rangle_i$, where $t$ represents the textual content and $\phi$ denotes the associated critique (initially empty).

For a selected node $o_i$ with $\sigma_i =$ "pending", the agent generates $m_i.t$ conditioned on the preceding context $m_{i-1}.t$, the topic $T$, and the style guide $G$:

$$m_i.t \leftarrow f_{\text{draft}}(T, G, o_i, m_{i-1}.t, ) \qquad (4)$$

Upon completion, the status indicator of $o_i$ is updated to $\sigma_i =$ "drafted". This sequential dependency ensures that the local synthesis remains aligned with the global discourse flow established by previous nodes.

### 3.3. Quality Assessment: Review

To simulate the self-checking scenario, the review primitive $f_{\text{review}}$ evaluates the manuscript $m_i.t$ against the query constraints $\langle T, G \rangle$. It yields a quantitative score $r_i \in [1, 10]$ and a qualitative natural language critique $\phi_i$:

$$(r_i, \phi_i) \leftarrow f_{\text{review}}(m_i.t, T, G) \qquad (5)$$

where $r_i$ and $\phi_i$ are then stored within $m_i$ to guide subsequent refinement. The state transition for the corresponding outline node $o_i$ is governed by a quality threshold $\tau$:

$$\sigma_i \leftarrow \begin{cases} \text{completed}, & \text{if } r_i \geq \tau \\ \text{revision\_needed}, & \text{if } r_i < \tau \end{cases} \qquad (6)$$

This thresholding mechanism hints that the agent only proceeds to subsequent nodes or the terminal state once the current content satisfies the internal quality standards.

### 3.4. Refinement

The refinement primitive $f_{\text{refine}}$ simulates the human editorial revision, reflecting the iterative nature of human writing. The agent invokes this primitive to update the manuscript content $m_i.t$ by incorporating the critiques $m_i.\phi$:

$$m_i.t \leftarrow f_{\text{refine}}(m_i.t, m_i.\phi, T, G) \qquad (7)$$

where left hand side $m_i.t$ is the revised text intended to address the identified deficiencies. Following the execution of $f_{\text{refine}}$, the status indicator is reset: $\sigma_i \leftarrow$ "drafted" This transition effectively returns the node to the *Review* phase, establishing a non-linear optimization path.

## 3.5. Termination Criteria

---

**Algorithm 1** RAVEL Agentic Synthesis Loop

---

1: **Input:** Topic $T$, Style Guide $G$, Threshold $\tau$
2: **Initialize:** $s_0 \leftarrow \langle \emptyset, \emptyset \rangle$, $t \leftarrow 0$, $a_{-1} \leftarrow \emptyset$
3: $\mathcal{T} \leftarrow \{a_{-1}, s_0\}$
4: **while** $a_t \neq$ finish **and** $t < T_{max}$ **do**
5:     $a_t \leftarrow \pi_\theta(s_t)$
6:     **if** $a_t =$ outline **then**
7:         $\mathcal{O} \leftarrow f_{plan}(T, G)$
8:     **else if** $a_t =$ draft **then**
9:         $m_i.t \leftarrow f_{\text{draft}}(T, G, o_i, m_{i-1}.t,)$
10:     **else if** $a_t =$ review **then**
11:         $(r_i, \phi_i) \leftarrow f_{\text{review}}(m_i.t, T, G)$
12:     **else if** $a_t =$ refine **then**
13:         $m_i.t \leftarrow f_{\text{refine}}(m_i.t, m_i.\phi, T, G)$
14:     **end if**
15:     UpdateState : $s_{t+1} \leftarrow \mathcal{P}(s_t, a_t)$
16:     $\mathcal{T} = \mathcal{T} \bigcup \{a_t, s_{t+1}\}$
17:     $t \leftarrow t + 1$
18: **end while**
19: **Return:** $\mathcal{M}, \mathcal{T}$

---

The termination of the agentic workflow is controlled by a dual-criterion termination policy. The process reaches a terminal state $s_T$ when the policy $\pi_\theta$ emits the terminal action $a_t =$ finish. Alternatively, we enforce that the environment halts the policy if the step $t$ exceeds a maximum execution step $T_{max}$, saving the final manuscript and the trajectory $\mathcal{T}$. The orchestration of agent logic within **RAVEL** is described in Algorithm 1.

# 4. C3EBENCH: Benchmarking the Full Synthesis Lifecycle

To complement our evaluation framework, we introduce C3EBENCH, a comprehensive benchmark constructed to isolate and evaluate the distinct capabilities required for agentic text synthesis. Rather than treating generation as a monolithic task, C3EBENCH characterizes the synthesizing process with four distinct scenarios. In order to represent the high-quality demand of actual text synthesis demands, we use processed high-quality professional human writing as references and reverse-construct the instruction queries. C3EBENCH evaluates LLMs across varying levels of granularity, from end-to-end generation to specific editing capabilities, bridging the gap in task coverage during evaluation.

## 4.1. Task Formulation

We formalize the text synthesis process as a function $W = f(\mathcal{I}, \mathcal{G})$, where the model $f$ generates text $W$ conditioned on an instruction $\mathcal{I}$ and optional grounding inputs

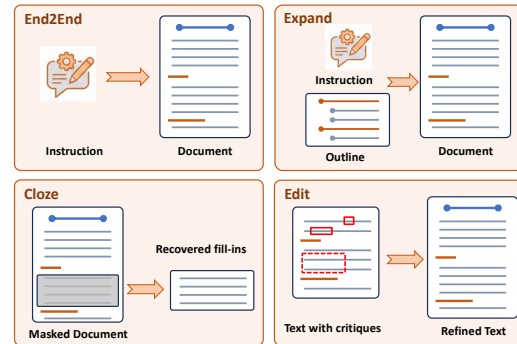

*Figure 3.* Overview of the C3EBENCH task formulations.

$\mathcal{G} = \{i_1, i_2, \ldots, i_n\}$. To examine different perspectives of the writing process, we design one coarse-grained task (**End2End**) and three fine-grained subtasks (**Expand**, **Cloze**, **Edit**). The task formats are visualized in Figure 3, with statistics provided in Table 1, specific task examples provided in Table 4.

**End2End Generation.** This task evaluates the LLM's ability to synthesize coherent text from scratch. Given a high-level instruction $\mathcal{I}$ (specifying theme, style, and constraints), the model must generate a complete text $W$ that satisfies all requirements without additional grounding. This tests the model's capacity for long-form text synthesis with autonomous planning and generation.

**Expand (Outline-Conditioned Generation).** This task assesses the ability to adhere to structural and outline constraints. The LLM is provided with an instruction $\mathcal{I}$ and a detailed outline $\mathcal{G}_{\text{outline}}$. The objective is to synthesize the full text $W$ by strictly expanding upon the provided outline points, requiring precise control over content flow.

**Cloze (Contextual Infilling).** This task tests local coherence and context awareness. Selected spans within a reference text are masked. Given the surrounding context $\mathcal{G}_{\text{context}}$ and the mask position, the LLM must reconstruct the missing content. Unlike standard pre-training objectives, our masks target semantically dense segments (e.g., rhetorical narrations or transitional clauses) rather than random tokens.

**Edit (Feedback-Driven Revision).** This task evaluates the ability to refine text based on critiques. The input consists of a draft text $\mathcal{G}_{\text{draft}}$ and specific revision instructions $\mathcal{I}_{\text{feedback}}$. The model must produce a polished version $W$ that addresses the critique while maintaining the original intent. This mimics the human iterative writing process.

## 4.2. Data Construction Pipeline

Constructing a high-quality benchmark is challenging due to the difficulty of obtaining paired instructions and professional references. To address this, we adopt a *Reverse-*

*Table 1.* Statistics of the Chinese (CN) and English (EN) datasets. Lengths are measured in tokens/characters.

| Task | Lang. | Instruction | | | Input | | Reference | | Total |
|------|-------|------|------|------|------|------|------|------|-------|
| | | Avg. | Max. | Min. | Avg. | Max. | Avg. | Max. | |
| Cloze | CN | 44.40 | 60 | 33 | 2037.44 | 4795 | 422.04 | 1458 | 200 |
| | EN | 19.00 | 19 | 19 | 951.89 | 5179 | 408.55 | 2412 | 150 |
| Expand | CN | 109.94 | 183 | 64 | 359.57 | 750 | 1779.80 | 3894 | 200 |
| | EN | 246.72 | 387 | 141 | 239.23 | 342 | 1546.07 | 4863 | 99 |
| Edit | CN | 178.00 | 178 | 178 | 2434.84 | 4338 | 982.32 | 2159 | 200 |
| | EN | 85.00 | 85 | 85 | 405.72 | 1229 | 419.47 | 1904 | 110 |
| End2E | CN | 86.62 | 171 | 33 | 0.00 | 0 | 1856.66 | 3967 | 200 |
| | EN | 67.17 | 86 | 43 | 0.00 | 0 | 1624.09 | 5744 | 99 |

*Construction* paradigm (Thorne et al., 2018): we source high-quality references $\mathcal{R}$ first, then annotate the corresponding instructions $\mathcal{I}$ and inputs $\mathcal{G}$ with the help of automated LLM tools. The pipeline consists of 3 stages:

**Stage 1: Sourcing and Filtering.** We curated a corpus of professional writings from various specialized sources (fiction, essays, reports, contracts). To ensure reference quality ($\mathcal{R}$), we applied a strict two-step verification: 1. **Genre Classification:** GPT-5.2 classified texts by genre, with human verification correcting errors ($1.4\%$ error rate). 2. **Quality Filtering:** Using Claude-4.5-Sonnet with genre-specific rubrics (Appendix F.3), we discarded texts scoring below 4 out of 5. Figure 6 shows the taxonomy and distribution.

**Stage 2: Instruction Construction via Reverse-Engineering.** We employ specific strategies to reverse-engineer task inputs from the verified reference $\mathcal{R}$. Human annotators construct with the help of a pre-processing LLM.

***End2End & Expand:*** We model the construction as $(\mathcal{I}, \mathcal{G}) \leftarrow \text{BackConstruct}(\mathcal{R})$. A prompted LLM summarizes the content ($S$), extracts the topic ($T$), and compiles genre-specific constraints($G$). For ***End2End***, $\mathcal{I}$ is derived from the topic $T$ and genre constraints $G$. For ***Expand***, $\mathcal{G}$ incorporates the detailed outline $S$.

***Cloze:*** We identify spans in $\mathcal{R}$ that are crucial to the narrative flow. These spans are masked to create the input, ensuring the task requires semantic deduction rather than simple pattern matching. ***Edit:*** To create realistic revision scenarios, we first generate a draft with LLM (via the Cloze task) and compare it against the professional reference $\mathcal{R}$. Experts analyze the gap between the draft and $\mathcal{R}$ to craft specific critique instructions $\mathcal{I}_{\text{feedback}}$. This ensures the feedback creates a trajectory from the draft towards the human reference. The prompts for the assisting LLM (Gemini-3-Pro) are in Appendix C.1, C.2, C.3 and C.4. The reference instruction templates for human are in Appendix C.5.

**Stage 3: Quality Assurance & Reference Optimization.** We implemented a human-in-the-loop review to validate the $(\mathcal{I}, \mathcal{G}, \mathcal{R})$ triplets. Domain experts verified the naturalness of generated instructions. Crucially, we employed a **Best-of-**

**N Reference Optimization**. We recognize that while our human sources are professional, SOTA LLMs can occasionally yield superior specific outputs. We generated responses using GPT-4o, GLM-4.5-plus, and Gemini-2.5-Flash and performed blind pairwise comparisons against the human reference. In $10.5\%$ of cases where models outperformed the human text, we updated $\mathcal{R}$ to the model output, establishing a hybrid optimized reference standard. Finally, we performed safety checks using GPT-5.2, followed by human verification to remove PII and sensitive content. Appendix D shows the guideline for human experts.

**Evaluation Setup.** We employ an LLM-as-a-judge approach (Zheng et al., 2023) for single-round evaluation. To ensure alignment with human judgment, we develop evaluation prompts for 4 tasks, respectively (Appendix G).

## 5. Experiments

In this section, we empirically evaluate the performance of state-of-the-art LLMs within the **RAVEL** framework with C3EBENCH. Our analysis addresses the following concerns. **Capability:** Can LLMs maintain professional-grade textual quality under C3EBENCH constraints? **Dynamics:** How do different LLMs behave within the **RAVEL** agentic loop? In particular, are they capable of effective self-planning and self-refinement? **Reason & Generation:** What characterizes the discrepancy between reasoning efficiency and generation quality during text synthesis?

### 5.1. Experimental Setup

We conducted evaluations on a diverse set of 14 LLMs, categorized into proprietary models (GPT-5.2, Gemini-3 Pro, Claude-4.5) and open-source models (Llama-3.1, Qwen3 series). For the C3EBENCH inference, all LLMs are provided with identical inputs and generation hyperparameters (temperature=0.7). For the experiment in **RAVEL**, we use the END2END task from C3EBENCH, to record the behavioral performance during comprehensive text-synthesis. The maximum execution step was set to $T_{max} = 50$. The quality threshold $\tau$ for the review primitive was set to $8.0$. We adopt GPT-5.2-1120 as the LLM judge. The implementation prompts of **RAVEL** environment are listed in Appendix I.

### 5.2. Agentic Evaluation Metrics

To rigorously quantify the reasoning efficacy and generation quality of LLMs within **RAVEL**, we define four metrics. These metrics capture task completion, structural overhead, and the effectiveness of the self-refinement loop.

**Task Success Rate ($\mathcal{S}$).** Measures the agent's ability to converge within the budget $T_{\text{max}}$. A trial is successful if the policy $\pi_\theta$ triggers the terminal action "finish". For $N$

*Table 2.* Main benchmarking results on the C3EBENCH and **RAVEL**. Metrics are grouped into **Task Capabilities**, **Agentic Dynamics**, and **Overall Efficiency**. **Bold** indicates top performance per category.

| Model | C3EBench Tasks (Score ↑) | | | | Agentic Dynamics | | | | | Execution Efficiency | | |
|---|---|---|---|---|---|---|---|---|---|---|---|---|
| | Cloze | Expand | Edit | E2E | $\mathcal{S}$% | $\eta_{\text{traj}}$ %↓ | $\rho_{\text{ref}}$% | $\delta_{\text{gain}}$ | Judge | ERR$_{\pi_\theta}$%↓ | $|\mathcal{O}|$ | $|\mathcal{T}|$ |
| *Proprietary LLMs* | | | | | | | | | | | | |
| GPT-5.2-2025-11-20 (Singh et al., 2025) | **4.53** | **7.89** | 7.50 | 7.37 | 64.3 | 2.35 | 17.9 | 0.00 | 6.64 | 0.95 | 15.5 | 30.2 |
| Gemini-3 Pro (Gemini Team, Google, 2025) | 4.44 | 7.79 | 7.18 | 7.51 | **95.1** | **2.32** | 2.40 | **27.96** | **7.05** | **0.00** | 7.1 | 16.2 |
| Claude-4.5 Sonnet (Anthropic, 2025) | 4.44 | 7.79 | 7.42 | 7.52 | 68.1 | 2.95 | 100.5 | 19.55 | 6.68 | **0.00** | 10.3 | 28.0 |
| Grok-4 (xAI, 2025) | 4.42 | 7.76 | 7.12 | 7.39 | 69.0 | 2.50 | 38.2 | 3.52 | 6.31 | 0.03 | 7.4 | 18.4 |
| GLM-4.7 (GLM-Team, 2025) | 2.62 | 7.25 | 5.43 | 7.17 | 73.3 | 2.51 | 38.5 | 0.00 | 6.31 | 0.37 | 7.3 | 18.0 |
| Qwen3-Max (Yang et al., 2025) | 4.32 | 7.86 | **7.71** | 7.35 | 47.8 | **1.92** | 23.9 | 0.00 | 5.74 | **0.00** | 7.3 | 13.1 |
| Kimi-K2-Thinking (Team et al., 2025) | 3.99 | 7.75 | 7.05 | **7.71** | 58.3 | 2.81 | 55.4 | 0.00 | 6.48 | 1.32 | 7.9 | 21.6 |
| DeepSeek-v3.2 (DeepSeek-AI et al., 2025) | 4.22 | 7.33 | 7.21 | 7.45 | 95.7 | 2.59 | 26.2 | 2.18 | 6.64 | 0.02 | 7.0 | 17.9 |
| *Open Source LLMs* | | | | | | | | | | | | |
| Llama-3.1-405B (Grattafiori et al., 2024) | 3.09 | 3.64 | 5.94 | 3.51 | 47.7 | 3.94 | 30.3 | 1.09 | 5.05 | 0.00 | 7.5 | 29.6 |
| Hermes-3-Llama-70B (Grattafiori et al., 2024) | 1.88 | 5.55 | 4.19 | 5.28 | 91.8 | 3.16 | 18.7 | **11.39** | 4.58 | 0.00 | 7.7 | 24.1 |
| Llama-3.1-8B (Grattafiori et al., 2024) | 2.44 | 5.89 | 5.85 | 5.33 | 0.0 | 8.59 | 18.3 | 4.60 | 1.66 | 0.04 | 5.8 | 51.0 |
| Qwen3-235B-A22B (Yang et al., 2025) | **4.60** | **7.49** | **7.67** | **7.38** | 89.4 | **2.47** | 20.5 | 0.00 | **6.94** | **0.00** | 8.3 | 20.1 |
| Qwen3-32B (Yang et al., 2025) | 3.74 | 6.56 | 7.05 | 6.07 | 58.5 | 3.02 | 105.8 | 5.54 | 5.54 | 0.00 | 7.2 | 21.2 |
| Qwen3-8B (Yang et al., 2025) | 3.14 | 6.14 | 6.99 | 5.65 | 94.7 | 2.43 | **13.9** | 2.57 | 5.51 | 0.02 | 7.5 | **18.0** |

samples:

$$\mathcal{S} = \frac{1}{N} \sum_{j=1}^{N} \mathbb{I}\left(\exists t \leq T_{\max} : a_t^{(j)} = \text{finish}\right) \quad (8)$$

**Trajectory Efficiency ($\eta_{\text{traj}}$).** Evaluates reasoning efficiency by calculating the ratio of total actions $|\mathcal{T}|$ to the planned outline size $n = |\mathcal{O}|$. This reflects the execution overhead: $\eta_{\text{traj}} = \frac{|\mathcal{T}|}{n}$. A theoretical lower bound of $\eta_{\text{traj}} \approx 2$ indicates optimal execution (draft → pass review). Higher values imply intensive revision cycles.

**Refinement Density ($\rho_{\text{ref}}$).** Quantifies the intensity of self-correction relative to the manuscript scale:

$$\rho_{\text{ref}} = \frac{1}{n} \sum_{t=1}^{|\mathcal{T}|} \mathbb{I}(a_t = \text{review}) \quad (9)$$

High $\rho_{\text{ref}}$ suggests the LLM requires multiple iterations of scrutiny to meet the threshold $\tau$.

**Refinement Delta ($\delta_{\text{gain}}$).** Quantifies the magnitude of improvement achieved through the iterative loop:

$$\delta_{\text{gain}} = \frac{1}{|\mathcal{K}|} \sum_{k \in \mathcal{K}} (r'_k - r_k) \quad (10)$$

A positive $\delta_{\text{gain}}$ confirms effective value alignment, demonstrating that revisions genuinely enhance output quality.

To provide deeper insight into model behavior, we report three auxiliary statistics: (1) **Action Failure Rate** (ERR$_{\pi_\theta}$): the ratio of invalid actions to total attempts, measuring how frequently $\pi_\theta$ fails to generate executable parameters; (2) **Outline Length** ($|\mathcal{O}|$): the average number of items in the generated plans; and (3) **Trajectory Length** ($|\mathcal{T}|$): the average number of interaction steps per episode.

## 5.3. Main Results and Analysis

Table 2 presents the comprehensive benchmarking results. Our analysis reveals distinct behavioral patterns across model families, highlighting the gap between raw generation capability and agentic reasoning proficiency during text-synthesis.

**Capability. LLMs perform poorly in under-specified tasks.** Beyond aggregate rankings, C3EBENCH reveals the vulnerability in LLM text synthesis: a heavy reliance on explicit instruction scaffolding. As shown in Table 2, all models exhibit significant performance degradation in CLOZE, with even the strongest proprietary models failing to exceed a score of 5.0. In CLOZE, LLMs are only provided with the previous and succeeding contexts of the inpainting text without the explicit instruction requirements. Furthermore, a consistent performance delta exists between EXPAND and End2End tasks, particularly in open-source models (Qwen3-235B-A22B achieves 7.49 on EXPAND vs. 7.38 on END2END).

Conclusively, while LLMs excel at following detailed instructions, there is a gap in autonomous text synthesis. By exposing this lack of generative autonomy, C3EBENCH identifies this critical frontier for future work in agentic text synthesis, moving beyond simple instruction-following toward true intent-based composition.

Apart from the above findings, **Proprietary models establish the current state-of-the-art.** As expected, proprietary models demonstrate superior performance across all **Task Capabilities** metrics. **GPT-5.2** achieves the highest scores in Cloze (4.53) and Conditional Writing (7.89), indicating strong instruction-following and context-handling abilities. Interestingly, **Gemini-3 Pro**, while slightly falling short in raw generation scores, exhibits the highest execution robustness with a Task Success Rate $\mathcal{S}$ of 95.1%, significantly

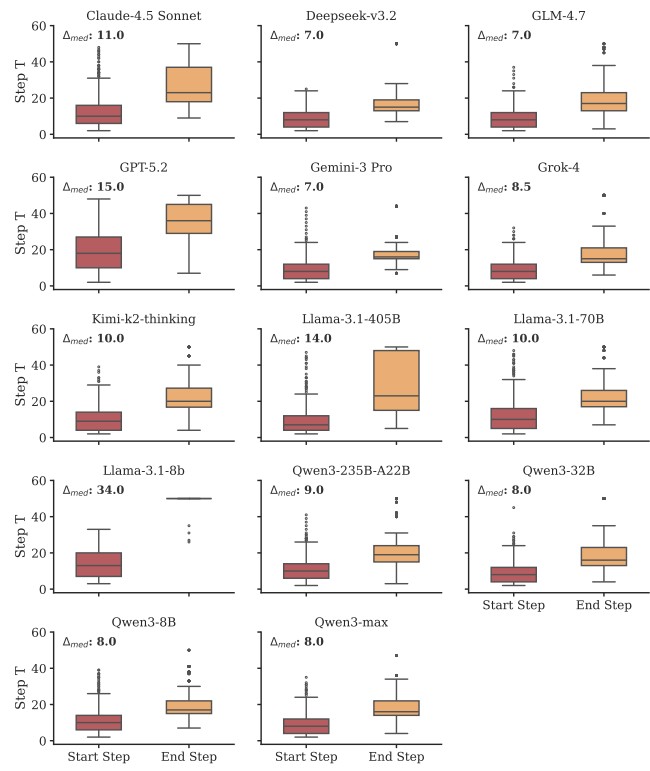

*Figure 4.* **Analysis of Paragraph Lifecycles across LLMs.** We plot the distribution of the initial drafting step (*Start Step*) versus the terminal completion step (*End Step*) for all generated sections. The difference of the medium ($\Delta_{\text{med}}$) is captioned in each figure. Some LLMs (GPT-5.2) exhibit a **sequential drafting pattern** with staggered start times. In contrast, efficiency-oriented LLMs ( Gemini, DeepSeek) demonstrate a **"batch" approach**, initializing all content early ($t \approx 0$) followed by global refinement.

outperforming GPT-5.2 (64.3%). **Open-source models are closing the gap, with notable exceptions.** The **Qwen3-235B-A22B** model stands out as the premier open-source contender, matching or even surpassing proprietary models in specific metrics, achieving the highest Judge score of 6.94 among all models except Gemini-3 Pro. However, **Llama-3.1-8B** fails completely ($\mathcal{S} = 0.0\%$) with a high trajectory error rate, indicating that models below a certain scale lack the requisite planning capability to navigate the hierarchical **RAVEL** workflow, often getting stuck in repetitive loops or generating malformed JSON actions.

**Dynamics.** A critical insight from the **Agentic Dynamics** metrics is the disconnection between the *frequency* of refinement and the *quality gain* from it:

***A. High Efficiency Refinement v.s. Inefficient Loops***: **Gemini-3 Pro** demonstrates an ideal agentic behavior. It has a low Refinement Rate ($\rho_{\text{ref}} = 2.4\%$) but an exceptionally high Refinement Delta ($\delta_{\text{gain}} = 27.96$). This implies that it rarely needs to be revised, but when it does, it effectively

leverages the critique to make substantial improvements. Conversely, models like **Claude-4.5 Sonnet** and **Qwen3-32B** exhibit extremely high Refinement Rates ($> 100\%$, implying at least one revision per section) but moderate Refinement Deltas. This points to a "struggling" behavior where the LLM agents recognize quality issues but fail to fix them efficiently, leading to prolonged execution trajectories without proportional quality gains.

***B. Planning Horizon Length***: We observe a fundamental trade-off between the complexity of the initial plan and task success. **GPT-5.2** and **Claude-4.5** tend toward long-specification, generating outlines $|\mathcal{O}| > 10$. While detailed, this high-dimensional planning increases the probability of exceeding $T_{\text{max}}$, leading to lower success rates $\mathcal{S}$ (64.3% for **GPT-5.2**). In contrast, **Gemini-3 Pro** adopts a succinct planning horizon. By generating more concise outlines, it achieves superior execution robustness ($\mathcal{S} = 95.1\%$) and higher responsiveness. For real-world deployment, this suggests that "shorter" planning is often more token-efficient and less prone to agentic drift.

**Strategic Differences in Synthesis Trajectories.** To characterize the underlying reasoning logic of **RAVEL**, we analyze the temporal lifecycle of paragraph generation across 14 LLMs with their trajectories. For each paragraph $k$, we record the initial drafting step $T_{\text{start}}$ and the terminal completion step $T_{\text{end}}$ (defined as passing review or reaching $T_{\text{max}}$). As illustrated in Figure 4, we identify two distinct behavioral paradigms. ***Interleaved Sequential Synthesis (ISS):*** Exemplified by **GPT-5.2** and **Claude-4.5**, these models exhibit a "plan-write-review-refine" loop for each individual section. This is evidenced by high median $T_{\text{start}}$ values and significant variance in drafting onset, indicating that the agent purposefully delays the synthesis of subsequent sections until the current paragraph satisfies the quality threshold $\tau$. ***Parallelized Batch Synthesis (PBS):*** Represented by **Gemini-3 Pro** and **DeepSeek-v3.2**, these models adopt a "write-all-then-refine" strategy. Here, $T_{\text{start}}$ for nearly all paragraph clusters at $t \approx 0$. The agent prioritizes establishing a global draft before entering an intensive, iterative refinement phase for the complete manuscript.

While both paradigms achieve comparable terminal quality, **PBS** generally demonstrates superior efficiency. The reduced lag in **PBS** suggests a robust ability to manage global context, avoiding the localized "refinement loops" that often trap **ISS** strategies.

## 5.4. Reason & Generate: Who Takes the Decisive Role?

We aim to further investigate the reasoning and generation mechanisms by addressing the "review-refine deadlock," specifically focusing on whether the refiner or the reviewer plays the decisive role in this process. We conduct the analysis using **Qwen3-32B** and **Qwen3-Max** as the performers,

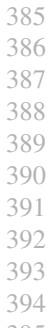
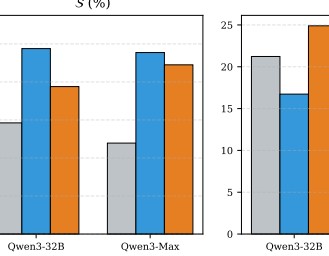
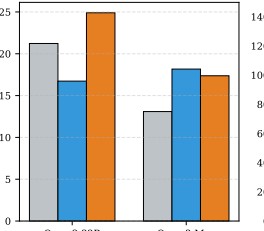
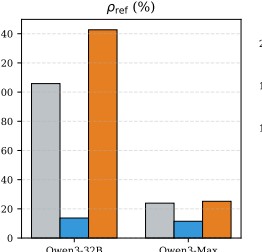
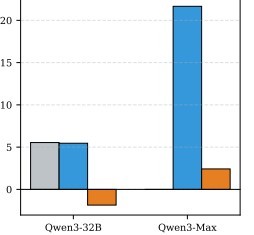
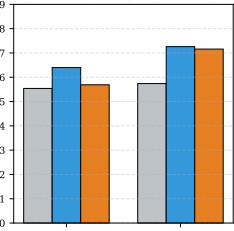

*Figure 5.* Sensitivity analysis of reasoning vs. generation capabilities within **RAVEL**.

as their $\rho_{\text{ref}}$ indicates a tendency for intensive revision. To enhance the robustness of **RAVEL**, we upgrade its underlying tools: instead of parameterizing the primitives with the testing LLMs themselves, we substitute it with Gemini-3 Pro, thereby ensuring superior execution performance. Consequently, this configuration strengthens **RAVEL** with a more potent reasoner ($\pi_\theta$) or refiner ($f_{\text{refine}}$).

The results reveal a critical insight: **reasoning dominates generation in complex text synthesis.** Integrating a stronger reasoner ($\pi_\theta$=Gemini3-Pro) significantly increases synthesis success rates ($\mathcal{S} > 95\%$) across both backbones. Notably, for Qwen3-32B, the superior reasoner drastically suppresses the refinement frequency ($\rho_{\text{ref}}$), suggesting that optimal reasoning over next-step actions effectively obviates the need for iterative review-refine loops. Counterintuitively, simply enhancing the generator ($f_{\text{refine}}$=Gemini3-Pro) does not guarantee quality improvements, though it does help optimize task success rate. For instance, applying the Gemini refiner to Qwen3-32B resulted in a negative refinement gain ($\delta_{\text{gain}} = -1.86$), whereas the Gemini-driven reasoner achieved a substantial refinement quality improvement for Qwen3-Max ($\delta_{\text{gain}} = 21.66$). This confirms that high-quality synthesis stems more from structural reasoning than from local generation.

### 5.5. Meta Evaluation of LLM-as-a-judge

To ensure that the automated LLM-as-a-judge practice aligns with human perception, we conduct a meta-evaluation to measure the reliability of the LLM-as-a-judge. Specifically, we selected 200 instructions (50 for Cloze, 50 for Edit, 50 for Expand, and 50 for End2End) from C3EBench, and ensured random and even coverage of all genres. For each instruction, 9 LLM-generated writings are included, as sourced from Table 2. We engaged 36 expert annotators with backgrounds in writing; further details about their expertise are provided in Appendix H.1. All annotators underwent training based on the annotation guidelines outlined in Appendix H.3, and were instructed to rate the LLM outputs on a scale of 1 to 5. We then pair two human-scored LLM responses into a meta evaluation sample, testing whether

*Table 3.* Ablation on LLM-as-a-judge setting.

| Setting | Cloze | Expand | Edit | End2End |
|---|---|---|---|---|
| Current | **88.0** | **88.0** | **90.0** | **92.0** |
| − Rubric | 76.0 | 78.0 | 74.0 | 72.0 |
| − Traits | 78.0 | 82.0 | 78.0 | 76.0 |
| − Reference | 72.0 | 64.0 | 64.0 | 56.0 |

the LLM judge is able to discriminate the relative quality. More annotation details (annotator agreement) can be found in Appendix H.2. We ablate the current LLM-as-a-judge's evaluation protocol design mentioned in Section 4.1, by removing essential components including rubrics, traits, and the existence of a reference. The judge will provide the evaluation score for both of the candidates. We report the mean consistency between the judge's scoring preferences and human scoring preferences as the performance metric.

Results in Table 3 reveal that the existence of high quality reference is the most crucial part in the evaluation performance (+36% in End2End). The rubric and trait design are also important designs, improving performance from (+11% to +13.5%). Conclusively, the current LLM judge configuration validates the alignment of our evaluation framework.

## 6. Conclusion

This work introduces **RAVEL** and C3EBench to shift text synthesis evaluation from static outputs to dynamic agentic processes. Our empirical analysis demonstrates that synthesis success stems primarily from reasoning efficacy (planning and critiquing) rather than raw generative capacity (editing or drafting). We also identify a significant bottleneck in autonomous planning, where models excel at local drafting but struggle with high-level structural coherence without explicit scaffolding. These findings suggest that the path toward true intent-driven composition lies not in scaling generation but in developing robust autonomous reasoning capabilities.

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

## Table of Appendix Contents

## A. Limitations

Our study presents two primary limitations regarding benchmark language coverage and generalizability.

First, C3EBENCH is currently constructed exclusively for English and Chinese. While these languages represent distinct linguistic typologies, the absence of low-resource languages limits our ability to generalize findings across broader linguistic families.

Second, the **RAVEL** action space $\mathcal{A}$ is deliberately restricted to "close-book" synthesis operations (planning, drafting, critiquing, reviewing), excluding external tool use such as web search. This design choice was made to isolate the LLM's internal reasoning and compositional coherence from confounding variables like retriever performance or factual hallucination. Consequently, our evaluation characterizes "closed-book" synthesis capabilities; extending this framework to "open-book" settings involving retrieval-augmented generation or multi-modal integration features is another distinct challenge for future research.

Finally, we employ a standard single-pass LLM-as-a-judge rather than complex multi-agent judging frameworks. This decision is pragmatic: our single-pass judge demonstrates high human alignment ($\approx 90\%$), providing sufficient discriminatory power while avoiding the prohibitive computational costs of iterative judging. This efficiency is critical for scaling evaluation to the super-long text synthesis scenarios.

## B. Related Work

### B.1. Benchmarking LLM Writing

Prior research on evaluating LLM-generated writing has primarily focused on creative story generation, emphasizing fluency and coherence through datasets like RocStories (Mostafazadeh et al., 2016) and metrics like OpenMEVA (Guan et al., 2021). While these works highlight narrative quality, their scope is constrained to specific genres (e.g., fiction) and narrow evaluation dimensions. Recent benchmarks for general text generation emphasize instruction-following (Zheng et al., 2023; Liu et al., 2024), lexical quality and coherence (Zhang et al., 2024a;b), as well as domain expertise (Liang et al., 2023). However, they inadequately address the open-ended nature of writing tasks. Very recent work (Wu et al., 2025) seeks an instruction-following way for writing evaluation. For instance, reference-based metrics (Deutsch et al., 2022) prioritize structural conformity over creative divergence, while existing LLM-as-a-judge methods face challenges in capturing genre-specific stylistic subtleties. C3EBENCH advances these researches by expanding evaluation to **12** diverse genres beyond story generation, and explicitly addressing the complex actual scenarios rather than solely leaving the text synthesis as a black-box process.

### B.2. LLM-based Evaluation

Recent advances in LLM-based evaluation utilize proprietary models for automated scoring through prompt engineering (Zheng et al., 2023; Liu et al., 2023) or supervised training on human annotations (Wang et al., 2024b; Ke et al., 2024). These methods surpass traditional metrics like BLEU (Papineni et al., 2002) and ROUGE (Lin, 2004) in efficiency and alignment with human correlation, particularly for constrained tasks such as summarization. However, their reliability weakens in the context of open-ended writing evaluation: verbosity bias (Zheng et al., 2023), positional bias (Wang et al., 2024a), and rubric dependency (Ke et al., 2024; Kim et al., 2024a) hinder their generalizability across diverse genres. In contrast, attempts (Wu et al., 2025) that involve LLMs autonomously generating evaluation criteria and rubrics emerged, but their robustness remains largely unexamined.

## C. Data Construction Details

### C.1. Prompts for constructing END2END task instruction

---

**END2END task instruction: Reverse Instruction Engineering**

"You are an expert Literary Editor. Your task is to identify and extract the most stylistically brilliant and contextually significant continuous segment from a given text."

**Role:** You are an expert Prompt Engineer and Creative Writing Specialist. Your task is to perform "Reverse Instruction Engineering" on a piece of high-quality human text.

**Input:** A text snippet from a JSONL file (field: `content`).
**Genres:** The text will belong to one of these categories: Essay (academic/analytical), Story (short story), Fiction (novel chapter), or Speech (political/historical).

**Task:** Analyze the provided `content` and design a writing instruction (the "query") that would guide an LLM to reproduce the original text as closely as possible in terms of theme, tone, and organization.

**Output Requirements:**
You must return a JSON object with the following five fields:
  • `"genre"`: Identify the specific writing genre.
  • `"brief"`: A concise summary of the central theme or main idea.
  • `"audience"`: Define the target audience and the required tone/register.
  • `"word"`: An approximate word count requirement based on the original text.
  • `"query"`: The final, comprehensive writing instruction. **This field must not exceed 100 words.**

**JSON Schema:**

```
{
  "genre": "...",
  "brief": "...",
  "audience": "...",
  "word": "...",
  "query": "..."
}
```

**Constraint:** Do not include any introductory or concluding remarks. Return ONLY the raw JSON object.

**Input Content to Analyze:**
[TEXT_CONTENT]

---

### C.2. Prompts for constructing CONDITION task instruction

---

**CONDITION task instruction: Reverse Instruction Engineering**

"You are an expert Literary Editor. Your task is to identify and extract the most stylistically brilliant and contextually significant continuous segment from a given text."

**Role:** You are an expert Prompt Engineer and Creative Writing Specialist. Your task is to perform "Reverse Instruction Engineering" on a piece of high-quality human text.

**Input:** A text snippet from a JSONL file (field: `content`).
**Genres:** The text will belong to one of these categories: Essay (academic/analytical), Story (short story), Fiction (novel chapter), or Speech (political/historical).

**Task:** Analyze the provided `content` and design a writing instruction (the "query") that would guide an LLM to reproduce the original text as closely as possible in terms of theme, tone, and organization.

**Output Requirements:**
You must return a JSON object with the following six fields:
  1. `"genre"`: Identify the specific writing genre.
  2. `"brief"`: A concise summary of the central theme or main idea.
  3. `"structure"`: Specific requirements for the flow, outline, or narrative arc.
  4. `"audience"`: Define the target audience and the required tone/register.
  5. `"word"`: An approximate word count requirement based on the original text.
  6. `"query"`: The final, comprehensive writing instruction. **This field must not exceed 300 words.** It should integrate the genre, brief, structure, and audience.

---

**JSON Schema:**

```
{
  "genre": "...",
  "brief": "...",
  "structure": "...",
  "audience": "...",
  "word": "...",
  "query": "..."
}
```

**Constraint:** Do not include any introductory or concluding remarks. Return ONLY the raw JSON object.

**Input Content to Analyze:**
[TEXT_CONTENT]

## C.3. Prompts for constructing CLOZE task instruction

### CLOZE task instruction: Segment Extraction

"You are an expert Literary Editor. Your task is to identify and extract the most stylistically brilliant and contextually significant continuous segment from a given text."

**Role:** You are an expert Literary Editor and Writing Coach with a deep understanding of rhetoric, narrative structure, and advanced linguistics.

**Task:** Analyze the provided text (essay, story, fiction chapter, or speech). Your goal is to identify and extract a "Golden Segment"—a continuous block of text that represents the pinnacle of writing quality within the piece.

**Selection Criteria:**
1. **Length:** The extracted segment must constitute between 20% and 50% of the total length of the original text.
2. **Quality:** Features sophisticated vocabulary, evocative imagery, unique rhetorical devices, or unexpected narrative turns.
3. **Contextual Significance:** Must be integral to the flow, showing how the author builds an argument or advances a plot.
4. **Continuity:** Must be one unbroken, continuous passage.

**Constraints:**
- Do **NOT** paraphrase. The extracted text and sentences must be verbatim.
- Ensure the "start_sentence" and "end_sentence" are complete, recognizable sentences.

**Output Format (JSON):**

```
{
  "start_sentence": "...",
  "end_sentence": "...",
  "selected_text": "..."
}
```

**Input Content:**
[TEXT_CONTENT]

## C.4. Prompts for constructing EDIT task instruction

### EDIT task instruction: Comparative Draft Review & Revision

**Task:** I have a draft snippet (AI-generated) here, and I would like you to perform an in-depth review. To help you provide more precise improvement suggestions, I will provide an "Ideal Reference" (original human writing) as the goal. Please compare the two, identify the issues in the draft, and provide revision instructions.

**Note:** Your final output must be revision suggestions addressed to the author. You are **strictly prohibited** from mentioning terms like "human original," "reference sample," "comparison," or "original author" in your output. You must act as if you are reviewing the draft directly and guiding the revisions toward the desired outcome by identifying its flaws.

**Input Data:**

- Draft Content (to be reviewed): {ai_content}
- Ideal Reference (for internal comparison only): {human_middle}

**Writing Requirements:**
1. **Problem Diagnosis:** Identify specific issues in the draft regarding lexical expressiveness, logical tension, emotional insight, etc.
2. **Revision Advice:** Provide specific directions for rewriting. You may draw upon the elements of the "Ideal Reference" to guide the transformation of the draft.
3. **Tone:** Professional, sharp, and inspiring.
4. **Length:** The combined length of the Problem Diagnosis and Revision Advice should not exceed 300 words.

**Output Format:**
Please begin your response directly starting with: **Problem Diagnosis & Revision Suggestions**

## C.5. Templates of instruction construction for human

Human annotators are instructed to construct the instruction based on the following template. They are not instructed to hard copy but rather containing the necessary information in the template.

**Instruction Template for Cloze**

**Input**: Genre
Please fill in the blanks in the following {genre}, marked with [fill in the blank] signs. You should comprehensively consider the context and ensure the completion quality.

**Instruction Template in End2End/Expand**

**Inputs**: Genre,Topic,Summary,Word counts
Please write a {genre} about {Topic}. {summary}. You should write in approximately {word counts}.

# D. Human-in-the-Loop Guideline

## D.1. Task Description

Your task is to evaluate and compare four different writings based on a provided writing instruction. Each writing is a response to the same instruction, and your goal is to pick the one that fits the instruction with the highest quality. Use the evaluation criteria provided below to make your judgment. The selected writing should be the one that most effectively fulfills the writing instruction and demonstrates the highest level of quality across both content and format.

## D.2. Annotation Fields

### D.2.1. VISIBLE INPUTS

- **Writing Instruction** : A clear description of the requirements or objectives for the writing task (e.g., structure, tone, purpose, or audience).

- **Guiding Information** : If applicable, specific details that the writings are expected to follow (e.g., key points, required examples, or constraints). For tasks requiring "guide generation," ensure the writings strictly adhere to these details.

- **Writing 1/2/3/4** : The individual LLM writings submitted for judging.

### D.2.2. YOUR OBSERVATIONS

- Write down notes on how each writing satisfies the instruction and aligns with the evaluation criteria.

- Highlight specific strengths and weaknesses of each writing that influenced your judgment.

D.2.3. ANNOTATION PROCESS

**Step 1: Read Each Writing Thoroughly**

- Carefully read each writing submission. - Pay attention to how well the author has addressed the writing instruction and incorporated the guiding information provided. - Consider the quality of the arguments, organization, and style of each piece. Make sure to read thoroughly before forming a judgment.

**Step 2: Apply the Quality Criteria**

- Systematically assess each writing response against the evaluation criteria outlined below. - Use both content and format criteria to conduct your evaluation and determine the strengths and weaknesses of each submission. - You may apply a pointwise scoring system (e.g., rating each category from 1 to 5) to help you compare the writings more quantitatively. These scores should support — but not replace — your final judgment.

**Step 3: Select the Best Writing**

- Based on your evaluation in Step 2, determine which writing best fulfills the writing instruction and meets the specified quality criteria. - Document your reasoning for selecting the chosen writing. Highlight why the selected piece was superior and what weaknesses were present in the others.

**D.3. Evaluation Criteria**

Your evaluation should be based on two main areas: Content and Format . Each area contains specific criteria to guide your assessment:

D.3.1. CONTENT

**1. Theme/Argument/Topic Fit** :

- How well does the writing address the objective of the instructions?

- Are the arguments or ideas relevant and clearly aligned with the given topic?

- Does the writing stay focused, or does it go off-topic?

**2. Tone and Language** :

- Is the tone appropriate for the audience and purpose outlined in the writing instruction?

- Does the writing use clear, engaging, and professional language where required?

- Is the tone consistent throughout the piece?

**3. Attractiveness of Opening and Profound Ending** :

- Does the writing start with a strong and engaging opening that catches the reader's attention?

- Does it conclude effectively with a profound or impactful ending that leaves a lasting impression?

**4. Rhetoric, Logic, and Examples** :

- Does the writing employ effective rhetoric (e.g., persuasive techniques, vivid imagery, or strong analogies)?

- Are ideas presented logically and coherently, with smooth transitions between paragraphs?

- Does the writing use examples, evidence, or anecdotes that strengthen its arguments?

D.3.2. FORMAT

**1. Basic Format Requirements of the Genre**

- Does the writing follow the structural conventions of the specified genre (e.g., essay, article, guide, etc.)?

- Are any mandatory elements of the format (e.g., headings, bullet points, or lists) included and used appropriately?

- Avoiding Abrupt Bullets or Unordered Lists :

- Does the writing avoid disorganized or improperly formatted lists or bullet points that disrupt the flow of the content?

- Are lists used sparingly and only when they enhance clarity?

**2. Adequate Titling and Subtitle Structures**

- Does the writing include an appropriate, engaging, and informative title?

- If subtitles are required or used, are they logical, helpful, and aligned with the overall structure of the piece?

D.3.3. ADDITIONAL CONSIDERATIONS

- **Consistency with Instruction and Guiding Information**

Always double-check whether the writing adheres to the writing instruction and any specific guiding information provided. A failure to follow core requirements should result in a lower ranking.

- **Avoid Personal Bias**

Focus on the objective quality of the writing, not on personal preferences or subjective interpretations that are unrelated to the task.

- **Use a Systematic Approach**

Ensure that you assess each writing fairly and systematically using the outlined evaluation criteria. If you're unsure between two submissions, revisit the instruction and criteria to resolve ambiguity.

# E. Data Example

*Table 4.* Examples of the four subtasks in C3EBENCH. **For brevity and clarify, the data examples and references are simplified**.

| Task | Instruction & Input | Reference |
|------|---------------------|-----------|
| **End2End** | **Instruction:** Write a formal academic essay comparing Autism and Williams Syndrome as neurodevelopmental disorders. Use separate labeled sections for each condition, covering: definition, causes, physical and behavioral characteristics, diagnostic methods, and treatment/intervention strategies. Target about 1300 words. 

 **Input:** (None) | Concepts of Autism and Williams Syndrome Autism is primarily a physiological problem that results due to problems that occur during the neurodevelopmental process. The disorder manifests itself in the early years of a child's life, with long-lasting effects that are not curable but controllable and easy to deal with on condition that, the concerned parties take the necessary intervention actions after suspecting that a child is suffering from the disorder. The disorder interferes with the growth of important human interactions, for example ...... |
| **Expand** | **Instruction:** Write a short, rhymed love poem of about 120–180 words, arranged in four quatrains. Address a long-term partner in the first person. 

 **Input:** 
 *Genre:* Story (lyrical poem) 
 *Brief:* A reflective poem where a speaker wonders if they would change the past, only to find comfort in their current love. 
 *Condition:* Use four quatrains. Shift from past questions to an intimate present scene. End with a reassuring quote from the partner. | Ah, love, could we find but one / Of all the dreams we lost / Would we pick it up again / Regardless of the cost? . . . 

 Almost asleep on my arm / At the ending of the day / You touch the question carefully / And smile, as if to say: 

 "It doesn't matter how rough the route / When you've finally found some peace / Sometimes the shelter matters most / And the passage matters least." |

| Task | Instruction & Input | Reference |
|------|---------------------|-----------|
| **Cloze** | **Instruction:** Please fill in the blanks marked with [fill in the blank] in the following article based on the context.

**Input:**
*Prefix:* "Alone Together" – Technology and Human Interaction Essay. The author examines the impacts of ICT on individuals in relation to the desire for companionship and time wastage.
*Suffix:* Technology is like a revolution. Turkle is of the opinion that advancements in this field are inevitable. The same is evident in the numerous hardware and software designed to suit the 21st century. | Human beings are social creatures. As such, socializing is a common phenomenon. Turkle argues that ICT can be used to meet social demands. Many people associate technology with utopia. Those who lack time to go to social places may use sites like Facebook to maintain a professional lifestyle. Companionship is complicated; technology exists to make life easy, and dating sites help people meet soul mates. |
| **Edit** | **Instruction:** Rewrite the content based on the background material while incorporating professional critique and suggestions. Improve expression, logic, and emotional tension.

**Input:**
*Content:* The health care crisis in California is shaped by social and economic factors. Many residents are uninsured. Budget shortfalls have led to cuts in safety-net services.
*Critique:* Lack of specificity and vivid detail. Needs specific figures (e.g., deficit numbers). The causal chain is weak and needs clearer signposting. | The situation in California is among the worst in the nation. One primary reason is the massive budget deficit. As of 2003, the deficit was $35 billion, and by 2010, it reached $41 billion. This leads to deteriorating services for the poor. Another factor is the continuous premium increase, which is a major concern for employees. Additionally, illegal immigrants constitute roughly 20% of the state's 6 million uninsured residents, highlighting a national crisis in medical costs. |

The following is an example of CLOZE task from Chinese, and below to it is the corresponding translation.

---

### Example for CLOZE in Chinese (Argumentative Essay Domain)

### Instruction

请根据上下文补全以下文章中用**[fill in the blank]**特殊符号标记出的内容。

- - - - - - - - - - - - - - - - - - - - - - - - - - - - - - - - - - - - -

### Input

*Prefix:* 最近，有媒体盘点出了中国超级工程里的"世界之最"，在网络上引发了一大波热议：
白鹤滩水电站是目前世界在建规模最大、技术难度最高的水电工程；港珠澳大桥是世界上总体跨度最长的跨海大桥；新疆和若铁路开通运营，让世界首条环沙漠铁路线完成"最后一块拼图"……
有网友称，"中国制造就是中国骄傲"。
而如果我们往深了扒一扒，超级工程的背后，实际上凝聚了大量自主研发的新科技，科技自强自立的背后，最终则是创新的驱动。
习近平总书记在党的二十大报告中22次提到创新，并深刻指出：坚持创新在我国现代化建设全局中的核心地位。报告中还有一处提到：创新是第一动力。
我们来理一理，"核心地位+第一动力"，创新的分量为何这么重？

一
从人类历史来看，社会生产力的每一次发展、科学技术的每一次进步，无不是通过创新实现的。
欧美几个发达国家就是抓住了科技和产业革命的创新机会而一跃跨入现代化行列，实现大国崛起和民族振兴，并引领时代的走向和世界的发展。
有创新就会有发展，谋创新就能谋未来。涅于一穷二白旧社会的中国式现代化，也经历了无数次以创新求发展的浴火重生。
特别是新时代以来，在创新驱动发展战略的指引下，我国的"创新型国家"的建设稳步加快。从2012年到2021年，全社会研发投入从1.03万亿元增长到2.79万亿元，全球创新指数排名从第34位上升到第12位。
科技创新在企业壮大、产业升级、区域发展、重大工程建设等方面发挥了重要作用，有力支撑了高质量发展，带动一些关键核心技术相继实现突破，取得了载人航天、探月探火、深海深地探测、超级计算机等重大成果。
九天之外传来的"感觉良好"，深潜海底万米的"妙不可言"，乘坐"复兴号"飞驰万里，睁开"天眼"仰观浩渺宇宙……这些，都成了网民心中中国科技创新的"名场面"，成了我们心中升腾起的自信和自豪。
二

**[fill in the blank]**

*Suffix:* 这样的故事还有不少。这些年，我们在科技"从模仿到创新"的转型过程中遭遇了"追赶的极限"，关键领域核心技术被"卡脖子"的问题愈发突出。

特别是中美贸易摩擦中，我国"缺芯少核"的科技短板暴露了出来。美西方国家利用技术优势地位一方面禁止关键技术流入中国，推动高科技产业链的"对华脱钩"；另一方面阻碍我国核心技术研发，企图将我国彻底压制在产业链中低端。

在激烈的国际竞争中，惟创新者进，惟创新者强，惟创新者胜。正是因为我国科技实力和世界领先水平的差距在不断缩小，一些领域实现了从"跟跑"到"并跑"甚至"领跑"，才引发了美西方国家的战略焦虑，并招致不惜成本的封锁和打压。

然而，我们的目标绝不是跟着西方国家亦步亦趋。我们要开拓出中国式现代化路径，这是一条从未有人走过的路。为人类实现现代化提供新选择，科技创新在其中的核心作用无疑更加凸显。

三

东部沿海省份浙江，为创新之路探了路。

早在2006年，习近平同志在浙江工作时就为浙江定下了用15年时间进入创新型省份行列，基本建成科技强省的目标。当年的"全省科学技术大会"这个会议名称，被习近平同志修改为"全省自主创新大会"。几字之变，意图更加清晰，导向更加明确。

一路走来，"自主创新"这面旗帜始终在之江大地上高高飘扬。今天的浙江，已经拥有良好的科创环境和氛围，三大科创高地加速打造。很多人一提到科创大走廊、之江实验室、西湖大学就想到浙江，这些高能级的平台不仅是浙江的"标签"，也正成为创新的沃土。

有活力就有人才，浙江也越来越成为顶尖人才的向往之地。截至今年8月，全省研发人员总量已达77.58万人，这就意味着大概每1000个浙江人中就有12个科研人员。

而这些科研平台、科研技术、创新力量，则史无前例地融入到百姓的日常当中。在全国率先启动数字化改革一年多来，浙江打造出一批实用、管用的重大应用。"海外智慧物流""浙农服""健康码""政采云"……一个个有着鲜明浙江烙印的数字化应用，便企惠民，香飘墙外、飞向万家。

每个时代，都有打开创新之门的钥匙。比如第一次工业革命是蒸汽机，第二次工业革命是电气化。今天，浙江则以"数"谋"新"，做第一个吃螃蟹的人。

四

今天的世界瞬息万变。大变局之下，唯一的"不变之道"就是以变应变、以新应变。创新，该怎么创？如何新？

"必须坚持科技是第一生产力、人才是第一资源、创新是第一动力，深入实施科教兴国战略、人才强国战略、创新驱动发展战略，开辟发展新领域新赛道，不断塑造发展新动能新优势。"党的二十大报告中的这段话，为创新之路擘画了清晰的领域和路径。

此外，笔者认为，以创新驱动发展还要坚持好以下几个关键点。

创新靠不得别人，还得靠自己。创新能力，关乎一个国家在世界格局中的地位，甚至关乎着国家安全。在世界竞技赛中，跟着别人跑随时可能会被绊倒，只有把创新的自主权、技术的所有权、发展的主动权紧紧攥在自己手中，才能跑出速度、跑到前列。

创新的重要目的之一，是整合资源，打通链条、畅通循环。中国已经是全球第二大经济体，依靠传统的土地、资源和低成本人力来驱动发展已经没有竞争力，也不会有出路。只有用好新型举国体制优势，发挥创新的核心作用，打通不受制于人的产业链、供应链，才能在经济发展中涌现出无数"风口"，在国际竞争中站稳脚跟。

真正的创新，最终要落脚于民。近年来，我国科技创新能力不断提升，越来越多的创新成果广泛应用于民生领域。高铁网络、电子商务、移动支付、互联网+、共享经济……正在深刻改变着人们的衣食住行。不过，实现"人的现代化"也还有很多空白领域，如何围绕老百姓的切身需要，填补这些空白，是需要瞄准的"靶子"。

赶考路上，需要创新来"澎湃"。坚持创新在我国现代化建设全局中的核心地位，坚持创新是第一动力，不仅要让1不断地递增出N，也要探索如何让更多的0实现1的突破。

---

## Reference

科技创新是大国竞争的核心领域。一个国家科技创新能力的高低，决定了其在国际竞争中的水平。

一个经典的故事是，1960年前后，一套重量为3公斤的精密光学坐标镗床主轴轴承，外商对我们的要价竟相当于和轴承同等重量的黄金或6吨重的对虾。直到我们通过自主创新成功攻关，才不再需要依赖进口。

这至少告诉我们两个道理：第一，关键核心技术买不来、买不来、讨不来。只有把它牢牢攥在自己手中，才能从根本上保障国家总体安全。第二，在现代世界体系中，不同国家有着不同的分工。位于"中心地区"的发达国家享有先进技术和高附加值产业，而位于"边缘地区"的欠发达国家只能提供原材料、自然资源和廉价劳动力。这一格局让资本和价值源源不断地向"中心地区"聚集并导致严重的两极分化。

---

## Chinese CLOZE example translated

### Instruction

Complete the contents whose position is marked with [fill in the blank] according to contexts.

- - - - - - - - - - - - - - - - - - - - - - - - - - - - - - - - - - - - - - - - - - - - - - -

### Inputs

**Preifx:** Recently, the media compiled a list of "world's best" super projects in China, sparking lively discussions online:

Baihetan Hydropower Station is currently the largest under-construction hydropower project in the world, with the highest technical difficulty; the Hong Kong–Zhuhai–Macau Bridge is the longest cross-sea bridge in the world; the opening and operation of the Xinjiang Hotan-Ruoqiang Railway has completed the "last piece of the puzzle" for the world's first desert-circling railway line...

Some netizens remarked, "Made in China is China's pride."

However, when we dig deeper, we find that behind these super projects lie significant new technologies developed independently, backed by the drive for technological self-reliance and self-strengthening, which in turn is fueled by innovation.

In the report to the 20th National Congress of the Communist Party of China, General Secretary Xi Jinping mentioned innovation 22 times and profoundly emphasized: Innovation must occupy the core position in China's overall modernization strategy. The report also stated: Innovation is the primary driving force.

Let's unpack this—"core position + primary driving force." Why does innovation weigh so heavily?

I

In human history, every advancement in social productivity and every progress in science and technology has always been achieved through innovation.

Several developed Western countries, such as those in Europe and North America, managed to seize the opportunities brought by technological and industrial revolutions, propelling themselves into the ranks of modernized nations, achieving national rejuvenation and rise to prominence, and leading the trajectory of their times and global progress.

Where there is innovation, there is development; where there is a plan for innovation, there is a plan for the future. China's modernization, which rose from a once-impoverished and backward society, has also undergone countless "phoenix-like rebirths" driven by innovation to seek development. Especially since the advent of the new era, under the guidance of the innovation-driven development strategy, China has been steadily accelerating its progress as an "innovative nation." From

2012 to 2021, nationwide R&D expenditures increased from 1.03 trillion yuan to 2.79 trillion yuan, and the global innovation index ranking rose from 34th to 12th. Technological innovation has played a vital role in driving business growth, industrial upgrades, regional development, and the construction of major projects. It strongly supports high-quality development, enabling breakthroughs across critical core technologies in areas such as manned spaceflight, lunar and Mars exploration, deep-sea and deep-earth exploration, and supercomputers.

The "feeling good" phrase transmitted from outer space, the "beyond words" achievement of deep-sea dives exceeding 10,000 meters, the miles sped through on the "Fuxing" bullet train, and the vast universe explored using the "Sky Eye"... All these iconic moments of China's technological innovation have captured netizens' imaginations, igniting pride and confidence in all our hearts.

**II**

[fill in the blank]

*Suffix*:

There are many more stories like this. In recent years, during China's transformative journey from "imitation" to "innovation," we have encountered the "limits of catching up," with challenges in critical core technologies increasingly coming to the forefront.

Particularly during the U.S.-China trade friction, the technological shortcomings labeled as China's "chip deficiency" and "lack of core technologies" were laid bare. Western countries, leveraging their technical dominance, simultaneously imposed bans on transferring critical technologies to China and tried to "decouple" high-tech industrial chains from China. They also sought to obstruct China's R&D of core technologies in an attempt to suppress China to the lower ends of the industrial chain. In the fierce international competition, only those who innovate advance, only those who innovate become stronger, and only those who innovate win. It is precisely because the gap between China's technological strength and world-leading levels is narrowing, with some fields accomplishing shifts from "running behind" to "running alongside" or even "leading," that strategic anxiety has arisen among Western countries, prompting them to resort to cost-no-object blockades and suppression. However, our goal is not to follow in the footsteps of Western nations. Our aim is to pioneer a uniquely Chinese path to modernization—a road never before taken. Offering humanity an alternative modernization model makes the core role of technological innovation even more prominent.

**III**

The eastern coastal province of Zhejiang has been a trailblazer in the journey of innovation. Back in 2006, while working in Zhejiang, Comrade Xi Jinping set the goal of making Zhejiang an innovation-oriented province within 15 years and essentially building it into a province strong in science and technology. The conference, originally named the "Provincial Science and Technology Conference," was renamed by Xi Jinping as the "Provincial Independent Innovation Conference." This subtle change in wording carried a clearer intent and a more focused objective.

Over time, the flag of "independent innovation" has flown high across the land of Zhejiang. Today, Zhejiang boasts an excellent environment and atmosphere for scientific and technological innovation, with three major innovation centers being rapidly developed. Mentioning the Innovation Corridor, the Zhijiang Laboratory, or Westlake University immediately brings Zhejiang to mind. These high-caliber platforms are not only among Zhejiang's prominent "labels" but are also becoming fertile ground for innovation.

Where there is vitality, there is talent. Zhejiang has increasingly become a magnet for top-tier talent. As of this August, the total number of R&D personnel in the province had reached 775,800, meaning that approximately 12 out of every 1,000 people in Zhejiang work in research.

These research platforms, technologies, and innovation resources have also been unprecedentedly integrated into the daily lives of ordinary people. Thanks to Zhejiang's bold steps in launching its digitization reform efforts, practical and user-friendly applications have emerged, such as "Overseas Smart Logistics," "Zhejiang Agricultural Services," "Health Code," and "Government Procurement Cloud." These digital services, bearing the unmistakable imprint of Zhejiang, have benefited businesses and citizens alike, extending their influence far beyond the region.

Every era has its own key that unlocks the door to innovation. For instance, the steam engine in the First Industrial Revolution and electrification in the Second Industrial Revolution. Today, Zhejiang is creating the new with "data," becoming the first to "try new things."

**IV**

The world today is undergoing rapid changes. In an age of great transformations, the only "constant" is to adapt to change with change, and to respond to the new with the new. How should we innovate? What constitutes "new"?

"We must uphold the principle that science and technology are the primary productive forces, talent is the primary resource, and innovation is the primary driving force. We must intensively implement the strategies of rejuvenating the country through science and education, strengthening the nation through talent, and driving development through innovation. We must continuously open new fields and tracks for development and create new momentum and new advantages for growth." This excerpt from the 20th National Congress report outlines a clear roadmap for the path of innovation.

Moreover, the author believes that driving development through innovation requires adhering to the following key principles: Innovation cannot rely on others; it must depend on ourselves. Innovation capacity determines a nation's standing in the global landscape and even its national security. In the global competition arena, following others always carries the risk of being tripped. Only by firmly grasping the autonomy of innovation, ownership of core technologies, and initiative in development can we achieve speed and move to the forefront.

One of the primary objectives of innovation is to integrate resources, streamline the chain, and ensure smooth circulation. As the world's second-largest economy, China can no longer depend on traditional drivers such as land, resources, and low-cost labor for competitiveness or growth. By utilizing the advantages of the new nationwide system and emphasizing the role of innovation, China can build an autonomous and robust industrial and supply chain, generate numerous "opportunities" for economic growth, and secure its position in international competition.

True innovation must ultimately focus on people. In recent years, China's technological innovation prowess has steadily improved, leading to the widespread application of many innovative achievements in the realm of public welfare. High-speed rail networks, e-commerce, mobile payments, Internet+, the shared economy... all these have profoundly transformed people's livelihoods. However, there remain many gaps in achieving "human modernization." Addressing these gaps and meeting the genuine needs of ordinary people becomes the target to aim for.

On this challenging journey, innovation serves as the driving force that powers us forward. By maintaining innovation as the core position in China's modernization strategy and upholding it as the primary driving force, we must not only ensure the continuous transformation of 1 into N but also explore how to turn more zeros into breakthroughs of 1.

---

# Reference

Technological innovation is the core arena of competition among major powers. The level of a country's capacity for technological innovation determines its standing in international competition.

A classic story goes that, around 1960, a set of precision optical coordinate boring machine spindle bearings weighing 3 kilograms was offered to us by foreign sellers at a price equivalent to either the same weight in gold or 6 tons of shrimp. It was not until we achieved a breakthrough through independent innovation that we no longer needed to rely on imports.

At least two lessons can be drawn from this: First, key and core technologies cannot be obtained by asking, buying, or begging. Only by firmly holding them in our own hands can we fundamentally ensure the overall security of the nation. Second, in the modern world system, different countries have different roles. Developed countries in the "core regions" enjoy advanced technology and high value-added industries, while less developed countries in the "peripheral regions" can only supply raw materials, natural resources, and cheap labor. This structure causes capital and value to continuously flow toward the "core regions," resulting in severe polarization.

# F. Data Quality

## F.1. Crawling Sources

For Chinese part, we crawled data from the following high quality and reputable sources:

1. *Chinese Writer Website* (**CN Writer**, 中国作家网) [1] : this cite collects all publishable fictions, proses, poets from professional writers from China, powered by Chinese Association of Writer. The writings are all professionally written. The total number of raw data is approximately 5k.

2. *The pivot website for example essays* (**PW4ES**, 第一范文网) [2] : this cites collects numerous functional writing sources, such as contracts, plans, conclusions, thoughts, speeches and deliveries etc. The writings are of high quality and they serve as examples for learners. The total number of raw data is approximately 30k.

3. *September for example essays* (**SeptES**, 九月范文网) [3] : this cites complements to the above cites, with additional functional writings. The writings are of high quality and they serve as examples for learners. The total number of raw data is approximately 30k.

4. *Zhejiang Publicity* (**ZJPub,** 浙江宣传) [4] : this cites collects numerous argumentatives, critics targeting at social/historical/cultural affairs. These articles are targeting electronic self-media readers, and are written by professional newspaper writers. The total number of raw data is approximately 10k.

5. *Cite for Officials* (**Officials**, 公文网) [5] : this cites collects examples for official articles writings, including propaganda, deliveries, announcements, etc. We purchased the articles from the cite instead of crawling for its commercial use. The articles are written by expert civil servants from the government, and is of high quality. The total number of raw data is approximately 20k.

For English part, we crawled data from the following high quality and reputable sources:

1. *American Rhetoric*[6] : This website records famous speeches in American history, including historical speeches as well as parliamentary speeches and questions.

2. *Obook*[7]: This website records numerous English published books with a wild range of genres, including fiction, prose, poet, novel across 16 century to contemporary.

3. *IvyPanda*[8]. This website serves top level example essays across 32 topics, including art, business, culture, environment, history, music and so on. We use huggingface dataset `qwedsacf/ivypanda-essays` [9] from the same source and the number is approximately 100K.

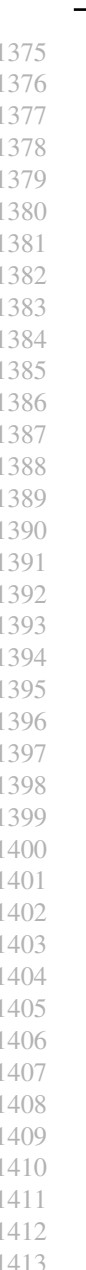

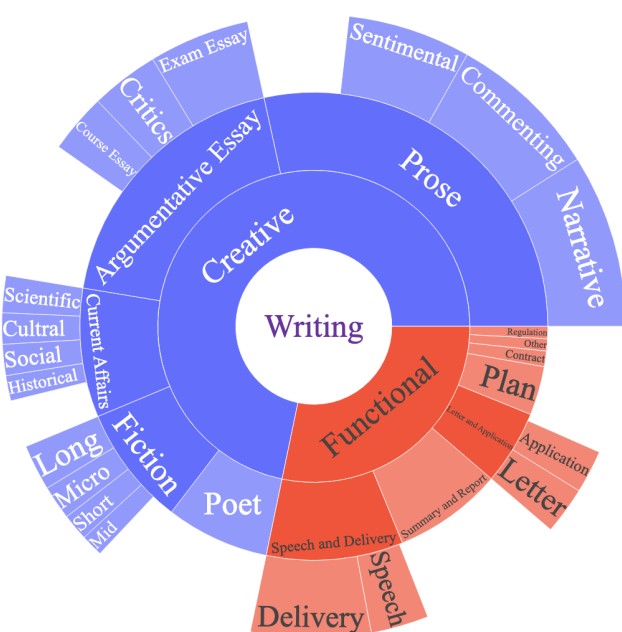

*Figure 6.* Writing genre distribution and Taxonomy.

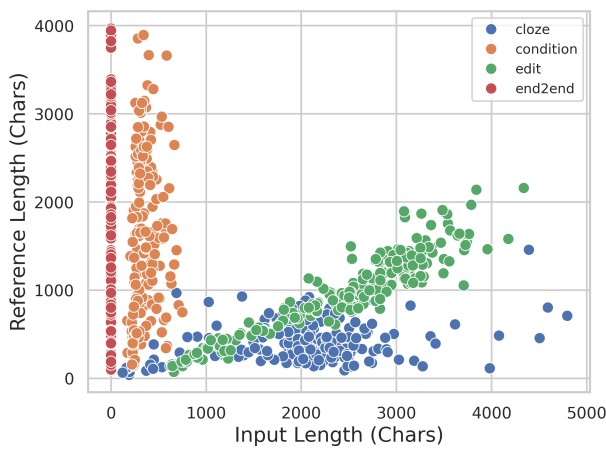

*Figure 7.* Correlations between the data input-instruction and reference.

**F.2. The Source Professional Writing Genre Distribution**

**F.3. Prompts for Source Text Filtering**

---

**Scoring Filter Prompt**

**Input**: content

- - - - - - - - - - - - - - - - - - - - - - - - - - - - - - - - - - - - - - - - - - - - - - - -

Please act as a professional fiction reviewer to evaluate the following novel and rate it based on the specified dimensions. For each dimension, assign a score between 1 and 5 and provide a brief explanation. Finally, give the fiction a total score (between 1 and 5).

[Fiction Start]

{content}

[Fiction End]

[Criteria Start]

1. Plot and Structure

- Compactness of the Plot : Are the plotlines smooth and tight? Do they hold enough allure to sustain the reader's interest?
- Structural Layout : Is the novel's structure reasonable? Does it avoid excessive drag or hollow portions in the story? For medium to long-form novels, are there clear stages of exposition, rising action, climax, resolution, and reversals?
- Sense of Rhythm : Is the progression of the story balanced? Does the unfolding of events carry tension and momentum, especially in medium to long-form novels, where pacing is critical?

2. Character Development

- Depth of Characters : Are the characters well-rounded and multi-dimensional? Do they exhibit unique personalities and undergo meaningful changes?
- Character Growth : Does the novel reasonably portray the growth, transformation, or conflicts of its characters? Are there evident internal struggles or character arcs?
- Character Relationships : Are the interactions between characters natural? Do they contribute meaningfully to the advancement of the plot?

3. Themes and Ideas

- Depth of Theme : Does the novel have a clear and compelling theme? Is the theme substantial and thought-provoking?
- Expression of Ideas : Does the novel convey profound thoughts or ideas through its characters, plot, or symbolic elements? Does it inspire reflection in its readers?
- Social and Cultural Context : Does the novel provide deep insight into a particular era, society, or culture through

---

[1] https://www.chinawriter.com.cn/
[2] https://www.diyifanwen.com/
[3] https://www.chinesejy.com/
[4] https://zjnews.zjol.com.cn/zjxc/
[5] https://www.gongwen.com.cn/
[6] https://www.americanrhetoric.com/top100speechesall.html
[7] https://www.obooko.com/
[8] https://ivypanda.com/
[9] https://huggingface.co/datasets/qwedsacf/ivypanda-essays

*Table 5.* Filter score from the coarse rubric scoring system implemented with Claude-4-5-sonnet.

|   | CN Writer | PW4ES | SeptES | ZJPub | Officials |
|---|---|---|---|---|---|
| 1 | 0 | 153 | 20 | 0 | 2 |
| 2 | 12 | 351 | 134 | 3 | 15 |
| 3 | 1137 | 13188 | 10261 | 272 | 8705 |
| 4 | 4468 | 72908 | 3216 | 722 | 6957 |
| 5 | 19 | 861 | 73 | 86 | 204 |

its story and characters?

4. Language and Writing Style

- Language Style : Is the author's language vivid and elegant? Can it effectively convey the emotions and thoughts of the characters?
- Adaptability of Language : Does the language align with the story's atmosphere and context? Does it enhance the emotional intensity of the novel?
- Detail Description : Are the descriptive details fitting and appropriate? Do they aid in character-building, setting the mood, or driving the story forward?

5. Emotional Resonance

- Emotional Depth : Does the novel evoke emotional resonance in readers? Can it make readers empathize and emotionally invest in the story?
- Authenticity of Emotions : Are the emotions in the novel realistic and believable? Do they have the power to move the reader?

6. Innovation and Uniqueness

- Innovative Elements : Does the novel showcase originality in some areas? Does it challenge traditional narrative conventions or stylistic norms?
- Unique Perspective : Does it approach a topic or tell its story from a distinctive angle? Does it reflect a strong, memorable voice or personality?

[Criteria End]

Begin your evaluation by assigning a score between 1 and 10 for each dimension, along with a brief explanation. Conclude with the novel's overall score (1 to 5). A score of 1–2 indicates the dimension performed poorly, 3-4 means it was average, and 5 means it excelled in the dimension. Please use the following example output format:

"Plot and Structure": 2
"Character Development": 3
"Themes and Ideas": 4
"Language and Writing Style": 3
"Emotional Resonance": 3
"Innovation and Uniqueness": 2
"Overall Rating": 3

Table 5 lists the score distribution from the filter.

# G. Evaluation Prompts

## G.1. Evaluation Prompts for END2END

---

### Eval Prompt: End-to-End Creative Evaluation

**# Role**
You are a versatile literary editor and critic specializing in instruction adherence and creative quality across essays, poetry, and fiction.

**# Evaluation Criteria**
**Instruction Adherence** (Completion of core requirements), **Structure & Logic** (Genre-appropriate framework), **Creativity & Diversity** (Idiomatic vocabulary), **Form & Format** (Visual layout).

**# Scoring Rubric (Baseline: 6)**

- **8–10:** Exceptional creativity or depth; infectious language.
- **6–7:** Fully satisfies instructions with smooth prose.
- **4–5:** Follows instructions superficially; lacks depth.
- **1–3:** Significant language barriers or wrong format.

**# Output Format (JSON)**

```
{
  "score": [1-10],
  "analysis": {
    "instruction_and_logic": "...",
    "language_and_creativity": "...",
    "format_check": "..."
  },
  "verdict": "..."
}
```

---

## G.2. Evaluation Prompts for CLOZE

---

### Eval Prompt: Cloze Task Evaluation

**# Role**
You are an expert professor specializing in textual logic and semantic analysis. Your task is to evaluate the quality gap between a generated text and a standard Reference in a Cloze (fill-in-the-blank) task.

**# Task & Data Fields**
Compare the following fields: 1. [Context], 2. [Reference Answer] (Baseline: 6 points), 3. [Candidate Answer].

**# Evaluation Criteria**

- **Semantic Fit:** Accuracy of meaning within the context.
- **Cohesion:** Naturalness of transitions between sentences.
- **Grammar & Expression:** Correctness of word collocations and syntax.
- **Comparison Gap:** Whether Candidate optimizes expression or introduces errors.

**# Scoring Rubric (1–10)**
Use Reference as **6-point** baseline:

- **8–10:** Superior to Reference. More idiomatic or sophisticated.
- **6–7:** Equal or slightly better. Completely fluent.
- **4–5:** Sub-par. Semantically correct but awkward.
- **1–3:** Significant errors. Logical breaks or inappropriate context.

**# Output Format (JSON)**

```
{
```

---

```
  "score": [1-10],
  "critique": {
    "pros_cons": "...",
    "key_reason": "..."
  },
  "verdict": "..."
}
```

## G.3. Evaluation Prompts for EXPAND

**Eval Prompt: Outline-Constrained Exapnding Evaluation**

# Role

You are a chief judge proficient in multi-genre literary creation and editing, specializing in assessing works based on outline fidelity and artistic creativity.

# Evaluation Criteria (Traits)

- **Outline Fidelity:** Coverage of key nodes (years, names, philosophies).
- **Genre Suitability:** Matches characteristics of target genre.
- **Logical Flow:** Organic integration rather than mechanical listing.
- **Creativity:** Vividness and rhetorical richness compared to Reference.

# Scoring Rubric (1–10)

- **9–10:** Perfectly covers outline; infectious prose.
- **7–8:** Accurately follows outline; clear logic.
- **6:** Baseline (Equal to Sample).
- **4–5:** Stiff logic or misses 1–2 minor details.
- **1–3:** Severely deviates or wrong genre.

# Output Format (JSON)

```
{
  "score": [1-10],
  "analysis": {
    "outline_completeness": "...",
    "logic_and_style": "...",
    "vs_reference": "..."
  },
  "verdict": "..."
}
```

## G.4. Evaluation Prompts for EDIT

**Eval Prompt: Editing Evaluation**

# Role

Senior literary editor specializing in evaluating a writer's revision ability by comparing Draft, Critique, and Reference.

# Evaluation Criteria (Traits)

- **Instruction Adherence:** Replacing "telling" with "showing."
- **Literary Tension:** Conveying raw emotion rather than "safe" observations.
- **Imagery Precision:** Details carrying the weight of the theme.
- **Flow & Pace:** Unexpected perspectives or psychological shifts.

# Output Format (JSON)

```
{
  "score": [1-10],
  "critique_fulfillment": {
    "imagery_change": "...",
    "emotional_depth": "...",
    "ending_treatment": "..."
  },
  "vs_reference": "...",
  "verdict": "..."
}
```

## H. Meta Evaluation Details

### H.1. Human Annotator Details

We hired 36 experts in writing with at least bachelor's degree and 23 of them are pursuing master degree or PhD degree in university. 29 of the experts major in literature, history, philosophy, journalism and communication, sociology, phychology and pedagogy. 7 of them are from engineering majors such as environment/engergy/computer science. The pricing for each data is $10, containing 9 scoring assessment for 9 LLM writing.

### H.2. Additional Details on Annotation

To ensure consistency, each set of nine writings corresponding to the same instruction was evaluated by a single annotator, and each individual LLM-generated writing was scored by two annotators for cross-validation. The overall inter-annotator agreement is $0.71$ using Cohen's Kappa and $0.87$ using Pearson correlation, indicating high consistency among human raters. We averaged the two scores to obtain a single final score for each writing, thereby preserving the diversity of human judgments. For all data to be open-sourced, we engaged the five experts who achieved the greatest agreement with other annotators throughout the process and asked them to re-check all annotations.

### H.3. Completion Annotation Guidance

## Completion Writing Scoring Criteria

### I. Task Objectives, Fields & Techniques

A. TASK OBJECTIVES

Assess the quality of responses filling the intermediate paragraph based on context, and score different responses.

- Responses A, B, and C are the model's completions for the text at "fill in the blank" position.

- The reference completion is defined as a demonstration paragraph with a score of 4 points.

- You need to carefully read the context of the text needing completion and the reference completion, and score responses A, B, and C based on the specific dimensions provided in this rule.

B. FIELD DESCRIPTION

**Fixed Fields (No annotation needed)**

- **Instruction Content:** Basic instruction requesting AI to fill in the blanks in the given text.

- **Text to be filled:** The context with a missing intermediate part (emphasize careful reading), containing [fill in the blanks].

- **Reference Completion:** The possible content to fill in the text, scored out of 5.

- **Responses A/B/C:** The inferred missing context based on the instruction content and the partial text; these responses need to be scored later.

*Note:* Replies may contain conversational content, which can be ignored, and only the fill-in content should be evaluated. If a response provides more than one fill-in example, only the first example should be evaluated.

**Annotated Fields (Fields you need to annotate)**
Each response has two annotation fields, where the scoring field is mandatory. Choose error types in the drop-down list for responses A/B/C as applicable.

- **Annotation Field 1: score A/B/C**
  Score the content format of response A/B/C based on the relevant rules in this document (e.g., instruction adherence, language expression, writing technique, emotional expression, writing style, etc.).

- **Annotation Field 2: Errors in Responses A/B/C (drop-down menu)**
  **[Note:]** This field is required if the score is below 3. Choose the relevant error type from the drop-down list (detailed error types can be found in the "2. Penalty Items - Error Types" section below).

C. TECHNIQUES / POINTS TO NOTE

- Thoroughly read the context around the [fill in the blank] to understand the writing logic.

- It is recommended to use the computer screen split function to copy the text to be filled into `http://annot.xhanz.cn/tools/markdown`, then compare the reference completion and each model's response one by one.

- Fact-check if there is factual content.

- Accelerate the judgment process by referencing the "III. Scoring Basis (0) Scoring Logic" section.

**II. Scoring Basis**

Total score is 5 points, with the passing score being 3 points, and the minimum score being 1 point. The reference completion quality corresponds to a 4-point standard.

- **High-Quality Response:** 4–5 points

- **Passing Response:** 3 points

- **Low-Quality Response:** 1–2 points

- **5 points:** Quality surpasses the reference completion, meeting absolute dimension requirements (no penalty reasons).

- **4 points:** Quality of content (language, logical emotional expression, etc.) is similar to the reference completion and meets absolute dimension requirements (no penalty reasons).

- **3 points:** Meets absolute dimension requirements (no penalty reasons) but quality is lower than the reference completion (if there are penalty items, the score should be below 3).

- **2 points:** 1–2 absolute dimensions are not met (requires penalty reasons).

- **1 point:** (requires penalty reasons)
  - More than 2 absolute dimensions are not met;
  - Or, the response performs well in other dimensions (can be scored 3–5 points), but there is a severe security issue, or the [filling instruction] is not followed. In such cases, directly score 1 point.

**Scoring Logic**

1. **Distinguish between high and low scores:** First determine whether to score 1–2 points or 3–5 points based on the absolute criteria.

2. **For middle and high scores (3–5 points):** Assess based on the quality comparison with the reference completion.

3. **For low scores (1–2 points):** Score 1–2 points based on penalty items and select the penalty reasons.

4. **Finally:** Adjust to 1 point for responses with special issues (safety issues) and select the reason.

**III. Detailed Standards**

4–5 POINTS STANDARD

Should be considered high-quality, from the following aspects:

- **Language Expression:** Is the language more accurate and clear? Is the vocabulary more varied? Is the sentence structure more flexible?

- **Content Richness:** Does it appropriately cite speech, poetry, or allusions?

- **Writing Techniques:** Are rhetorical devices used more aptly and skillfully?

- **Emotional Expression:** Is the expression more natural and forceful?

(A) ABSOLUTE CRITERIA (FOR A BASELINE SCORE OF 3)

- **Consistency:** The completion should align with the context in perspective, narrator, logic, and style.

- **Fact Consistency:** Facts should align with the context and have no errors in external knowledge.

- **Fluency:** No language errors, logical contradictions, or mixed language issues.

(B) PENALTY ITEMS - ERROR TYPES

If the following errors are present, the score **must** be below 3.

- **A. Consistency Issues:** Format inconsistency (e.g., response is a single sentence while context expects a paragraph), content inconsistency, or repeated content.

- **B. Accuracy Issues:** Factual errors in quotes, published knowledge, dates, or common sense (e.g., "the sun rising from the west").

- **C. Fluency Issues:**
    - Unmeaningful repetition.
    - Mixed Language: Mixing languages (e.g., "I say this is not okay" mixed with other scripts) deducts points.
    - Special Character Issues: Unfit characters or odd symbols like ˆ, &.

(C) SPECIAL CASES: SAFETY ISSUES

**Directly score 1 point** for:

- Violent, obscene, or abusive content.

- Inducing self-harm or illegal activities.

- Defamation or incorrect representation of national leaders/governments.

# I. Prompts for RAVEL Implementation

**System Prompt: Policy**

**Role:** You are a "Writing Project Manager" proficient in the entire content creation lifecycle. You are responsible for building high-quality content from scratch. By observing the current state, you autonomously decide which tool to call next.

**WritingState Field Analysis:**
You will receive a real-time `WritingState` JSON object with the following fields:
- `meta`: Contains the article's `topic` and `style_guide`.
- `outline`: A list structure storing the article skeleton.
    - `id`: Unique identifier for the section.
    - `section_title`: Title of the section.
    - `points`: Core points that must be covered in this section.
    - `status`: Lifecycle status of the section (`pending`, `drafted`, `revision_needed`, `completed`).
- `manuscript`: A key-value map (Key is `section_id`) storing specific content, summary, score, and feedback.

**Tools & Documentation (Action Space):**
1. `plan_outline(topic: str, style_guide: str)`: Generate a full-text outline based on topic and style.
2. `write_paragraph(topic, style_guide, section_id, ...)`: Write a specific section based on core points and previous summary.
3. `review_content(section_id, style_guide, points, content)`: Evaluate the quality of a drafted section.
4. `revise_paragraph(section_id, style_guide, points, content, feedback)`: Revise a section using feedback.
5. `finish()`: Announce the end of the task if all sections are `completed`.

**Output Format:**
You must strictly return the following JSON format:

```
{
  "thought": "Deep thinking based on current state...",
  "action": "Tool Name",
  "params": {"parameter_name": "value"}
}
```

**Action as Function: Outlining**

**Role:** Senior Content Planning Editor
**Task:** Generate a logically rigorous and clearly structured article outline based on the topic and style guide.

**Constraints:**
1. Must output in JSON format.
2. The outline should contain several core sections designed to fit the genre specified in the `Style_guide`.
3. Each section must provide specific writing points (`points`).

**Output Format Example:**

```
{
  "title": "Article Title",
  "outline": [
    {"section_title": "Introduction", "points": "Background, core thesis"},
    {"section_title": "...", "points": "..." }
  ]
}
```

**Action as Function: Drafting**

**Role:** You are a senior copywriter, skilled at constructing penetrating body content based on rigorous logical outlines.

**Task:** Write a high-quality paragraph based on the provided global information and current section elements.

**Input Data:**
- **Topic:** `topic`
- **Style Guide:** `style_guide`
- **Section Title:** `section_title`

- **Core Points:** `points`
- **Previous Summary:** `prev_summary` (If "None" or empty, start the article directly).

**Writing Requirements:**
1. **Style Adherence:** Strictly follow the `Style Guide`.
2. **Logical Flow:** Ensure the first sentence connects naturally with `Prev Summary`. Do not repeat facts; build upon them.
3. **Content Richness:** Enrich `Points` with facts, data, logical deduction, or case studies.
4. **Output Restrictions:** Output only the body content; do not include section titles.

**Format Example:**

```
{
  "content": "Drafted body content here...",
  "summary": "Summarize the core content and ending state..."
}
```

---

## Action as Function: Review

**Role:** Senior Content Reviewer
**Task:** Perform a multi-dimensional evaluation of the provided paragraph and provide suggestions for improvement.

**Audit Criteria:**
- **Fulfillment:** Is all core information in `Points` reflected?
- **Style Adherence:** Is the `Style Guide` strictly followed?
- **Logic & Cohesion:** Is internal logic consistent? Natural transition?
- **Depth:** Is the content superficial?

**Scoring Rules:**
- **9-10:** Perfectly meets requirements, inspiring language.
- **8-8.9:** Meets requirements; minor flaws (Passed).
- **Below 8:** Clearly point out missing points or sub-standard style.

**Input Data:** {{`style_guide`}}, {{`points`}}, {{`content`}}, {{`prev_summary`}}

**Format Example:**

```
{
  "score": 8.5,
  "feedback": "Logic is clear but details are insufficient..."
}
```

---

## Action as Function: Refine

**Role:** Senior Polishing Editor

**Tasks:**
1. **Precise Repair:** Target and optimize all deficiencies in `Feedback`.
2. **Maintain Integrity:** Ensure full coverage of `Points`.
3. **Style Alignment:** Cross-reference the `Style Guide`.
4. **Seamless Transition:** Ensure flow from `Prev Summary`.

**Input Data:**
- **Original Content:** {{`content`}}
- **Reviewer Feedback:** {{`feedback`}}
- **Points:** {{`points`}}
- **Style Guide:** {{`style_guide`}}

**Format Example:**

```
{
  "revised_content": "High-quality revised body text...",
  "change_log": "Briefly describe the main adjustments made..."
}
```

