# OpenReview forum: "RAVEL: Reasoning Agents for Validating and Evaluating LLM Text Synthesis"
_ICML.cc/2026/Conference — Submitted to ICML 2026_

### Official Review · Reviewer_Lxvm · 2026-03-09

**Soundness:** 2
**Presentation:** 2
**Significance:** 3
**Originality:** 3
**Overall Recommendation:** 3
**Confidence:** 4

**Summary:**

A pressing problem examined by this paper is how to evaluate long-form text synthesis in a way that reflects the actual multi-step writing process rather than treating writing as a single-pass generation problem. This submission claims to consider a general domain, namely text synthesis across multiple genres and writing settings, and it proposes to do so through two components: RAVEL, an agentic evaluation framework in which the tested LLM plans, drafts, reviews, and refines its own writing trajectory, and C3EBENCH, a benchmark of 1,258 English and Chinese samples spanning four synthesis scenarios: Cloze, Edit, Expand, and End-to-End. The paper argues that standard evaluations miss important differences in planning, self-critique, and revision behavior, and that evaluating full trajectories gives a better view of synthesis capability than judging final outputs alone.

The core technical setup formalizes text synthesis as a sequential decision process in which the model maintains an evolving state consisting of outline nodes and manuscript nodes, chooses from actions such as outline, draft, review, refine, and finish, and uses an internal reward signal to decide when quality is sufficient. The benchmark is built by reverse-constructing instructions and inputs from professional human references, then using human-in-the-loop validation and an LLM-as-a-judge setup for evaluation. The experiments compare 14 models and analyze both static benchmark scores and trajectory-level behavior inside the RAVEL loop. The paper’s main claim is that synthesis success depends more on reasoning ability, especially planning and critique, than on raw generation quality, and that stronger reasoners can guide weaker generators more effectively than the reverse.

**Compliance With Llm Reviewing Policy:**

Affirmed.

**Final Justification:**

I appreciate the additional clarification on hybrid references, judge validation, claim calibration, and the scope of the dynamic RAVEL experiments. In particular, the new detail on text-level versus system-level reliability makes the paper’s evaluation story more transparent.

However, I am maintaining my Weak Reject recommendation. My main concern remains that the evaluation continues to depend substantially on LLM-based judging, with weaker reliability at the individual-instance level than at the system-ranking level. In addition, I still view the “reasoning dominates generation” result as a promising but bounded empirical finding within this specific setup, rather than a general conclusion. Overall, the paper has clear merit, but I still think it would benefit from tighter scoping and stronger validation before acceptance, so I am keeping my score at 3.

**Key Questions For Authors:**

1. How should readers interpret the 10.5 percent of cases where human references were replaced by model outputs? Does this change the target from professional human writing to a hybrid reference standard, and how might that affect evaluation validity?

2. Why should the reader view the claim that reasoning dominates generation as established, rather than as a narrower observation from two Qwen-based case studies with Gemini substituted into selected roles?

3. Can the authors provide stronger validation for GPT-5.2 as judge beyond pairwise preference consistency, especially for absolute score reliability across the four tasks?

4. Since the dynamic RAVEL experiments are conducted only on the End-to-End task, how far do the authors believe the trajectory-level conclusions generalize to Edit, Expand, and Cloze?

5. In Algorithm 1, can the authors clarify the mismatch between the action names and primitive names, for example outline versus fplan, and explain more precisely how invalid actions and malformed parameters affect the transition function and success metric?

**Limitations:**

No. The paper does include a limitations section in the appendix, and it mentions language coverage, the closed-book action space, and the use of a single-pass judge. That is a good start. However, I do not think it adequately discusses some of the most important practical limitations of the current setup, especially the hybrid reference optimization procedure, the reliance on LLM-assisted reverse construction, the fact that only one task is used for the dynamic agent analysis, and the degree to which the conclusions depend on GPT-5.2 judging. I would encourage the authors to discuss those issues more openly in the main paper.

**Strengths And Weaknesses:**

Strengths:
--------------
1. The paper is asking a worthwhile question. The motivation is easy to appreciate: real writing is iterative, and many current evaluations collapse planning, drafting, reviewing, and revising into a single final-output score. That criticism of standard evaluation is reasonable and well presented.

2. The separation between the framework and the benchmark is sensible. RAVEL is not just another benchmark table. It is an attempt to expose synthesis trajectories, while C3EBENCH provides four task settings that map onto different parts of the writing process. That overall structure is coherent.

3. The benchmark construction pipeline shows real effort. The authors do not simply generate synthetic prompts from scratch. They begin from professional reference texts, reverse-construct instructions and grounding inputs, and include human verification, safety screening, and some quality control. That gives the dataset more thoughtfulness than a fully automated benchmark.

4. The benchmark covers multiple task granularities. End-to-End, Expand, Cloze, and Edit are meaningfully different, and this helps the paper say something more specific than a single overall synthesis score. In particular, the contrast between more scaffolded tasks and less specified tasks is a useful design choice.

5. Some of the empirical findings are interesting. The paper reports that models are much weaker on Cloze than on the more specified tasks, and the trajectory analysis in RAVEL does reveal different styles of behavior, such as more sequential drafting versus earlier global drafting followed by refinement. Even if I am not fully convinced by all of the interpretation, the paper does surface patterns that are more informative than a single leaderboard ranking.

6. The meta-evaluation of the judge is a positive inclusion. The authors at least try to validate their LLM-as-a-judge design with human annotations and ablations over rubrics, traits, and reference availability. Many papers would have skipped that entirely.



Weaknesses:
------------------
1. My biggest concern is that the paper makes fairly strong claims about evaluating synthesis operations, but in practice a large part of the setup still relies on self-referential LLM-based machinery. The tested model evaluates itself inside the loop through review and termination logic, while GPT-5.2 is also used as the external judge for the benchmark. That creates a somewhat circular evaluation story. The paper is trying to assess planning and refinement quality, but much of the signal is still generated by LLM-based critics rather than grounded in an external notion of writing quality.

2. The formulation is more suggestive than rigorous. The paper casts the process as a sequential decision process with an optimization objective, but this formalization does not really buy much beyond notation. The reward is self-estimated by the model, the threshold is fixed manually, and the decision process is not learned or analyzed in a principled way. As written, the formalization risks sounding more technically grounded than the implementation really is.

3. The benchmark design raises realism questions. The paper emphasizes that C3EBENCH is derived from professional human writings, but the actual instructions, outlines, critiques, and some task inputs are reverse-constructed with LLM assistance. In Edit, the draft is first generated by an LLM and then experts write critiques relative to the reference. In addition, 10.5 percent of human references are replaced with model outputs after blind comparison. That means the benchmark is not purely human-grounded in the way the framing initially suggests. I do not think this invalidates the benchmark, but it should be discussed much more directly as a limitation.

4. The reference optimization step is potentially problematic. Replacing human references with model outputs in 10.5 percent of cases may improve surface quality, but it also blurs what the benchmark is actually measuring. It becomes less clear whether the target is professional human writing, a model-preferred style, or a hybrid standard. This matters because the paper’s evaluation claims are about professional-grade text synthesis.

5. The main benchmark is not especially large for the breadth of claims being made. With 1,258 samples across four tasks, two languages, and multiple genres, the effective data per slice becomes modest. That is enough for an exploratory benchmark, but I am not convinced it fully supports broad claims about text synthesis capability in a general domain.

6. The use of GPT-5.2 as the sole main judge remains a weak point despite the meta-evaluation. The reported alignment numbers in Table 3 are encouraging, but they are framed as pairwise consistency with human preferences on selected samples rather than a full validation of absolute scoring quality. Also, the paper’s own appendix acknowledges that it uses a pragmatic single-pass judge rather than a richer evaluation setup. For a paper whose conclusions rely heavily on judged writing quality, I would have liked a stronger validation story.

7. The central claim that reasoning dominates generation is not yet fully established. The ablation in Section 5.4 is interesting, but it swaps in Gemini-3 Pro as a stronger operator for either the reasoning policy or the refiner on only two backbones that already exhibit high revision rates. That is a fairly narrow experimental basis for a broad claim about the decisive role of reasoning over generation in text synthesis. I would view this as a promising observation rather than a firm conclusion.

8. Some trajectory metrics are difficult to interpret as quality indicators. For example, high refinement density may indicate careful revision, poor initial drafts, poor reviewing, or unproductive loops. Similarly, success is defined by reaching a finish action within the budget rather than by independent confirmation that the output is truly good. The paper interprets these metrics confidently, but several of them are more ambiguous than the writing suggests.

9. The paper uses only the End-to-End task for the agentic RAVEL experiments. That is understandable for tractability, but it weakens the claim that the framework broadly evaluates the full synthesis lifecycle. If the dynamic conclusions are drawn only from one task type, the scope of those conclusions should be narrower.

10. Presentation is a noticeable weakness. There are quite a few grammatical problems, awkward phrases, and notation issues in the main paper. Some examples include “tasks that demanding contextual understanding” in the abstract, “that pairing text references” in the benchmark description, “critical irrelevancy” in the results section, “The difference of the medium” in Figure 4, and the inconsistent use of action names such as outline versus fplan in Algorithm 1. These are not minor copyedits only; in places they make the argument harder to trust.

11. Quick Question: Figure 4 caption on page 7 says “The difference of the medium”. Did you mean “median”?

---

> ### Author Rebuttal · Authors · 2026-03-31
>
> We appreciate your constructive feedback. Due to restricted space, we mainly respond to the key questions. Please do not hesitate to reach further questions or offering futher clarification chances.
>
> ## **LLM assistance**
>
> > Q1: Hybrid reference (also W3/W4)
>
> In our evaluations, these LLM outputs objectively surpassed the original human references in **formatting, redundancy, and internal logic**, which otherwise would have compromised evaluation accuracy. Logic behind this:
>
> - Human experts are aligned with the **same quality guideline** (Appendix D, IAA=0.87)
> - The ranked results are calibrated with that written standard.
> - Retaining flawed references artificially lowers the threshold, leading to inflated scores and compromised validity.
>
> To further address this concern, we discarded the 10.5% hybrid samples and re-calculated all metrics on the left samples (89.5%). The shifts does not change the drawing conclusions, which demonstrates the the quality/effectiveness is not influenced by the 10.5% part. Core metrics digest:
>
> **Capability**
>
> - Cloze: GPT-5.2 (4.53 -> 4.52);  Gemini-3 Pro - Cloze (4.44 -> 4.46)
>
> - Qwen3-235B: Expand (7.49->7.51); End2End (7.38->7.39)
>
> **Dynamics**
>
> - Success rate: GPT-5.2 (64.3 -> 64.1); Gemini-3 Pro (95.1 -> 94.9); Llama-3.1-8B:  0->0
> - Refine density: Claude-4.5 Sonnet (100.5->102.2); Qwen3-32B (105.8->103.6)
>
> ## **Claim boundary**
>
> > Q2 & W7: "Reasoning dominates generation" not solid
> >
>
> To validate beyond the initial Qwen+Gemini setup, we tested **diverse base models (Llama-3.1-70B, GLM-4.7) and augmentors (GPT-5.2, Claude-4.5).** As detailed in **our rebuttal to Reviewer Zmzi**, these comprehensive results consistently confirm that upgrading the reasoner yields significantly higher success rates and quality gains than upgrading the generator.
>
> We agree such claims require prudence. We will revise the text to discuss model-specific divergences (e.g., unlike GPT/Claude, Gemini struggles to assist weak bases as a refiner) to properly bound our conclusions.
>
> > Q4 & W9: comments on RAVEL's generalizability
>
> As you have pointed out, our trajectory-level conclusions strictly apply to End-to-End synthesis. For tractability, RAVEL initializes from an empty state. Evaluating intermediate tasks dynamically requires an **interrupt-and-restore mechanism** to parse static texts into structured agent states (outlines, partial drafts, critiques).
>
> We will explicitly narrow our claims in the limitations section. We are actively extending RAVEL's the experiments on the 3 scenes. Our comments are:
>
> - The extension shapes the three tasks into long-horizon agentic tasks, where the reasoning dynamics are similar in E2E (starting from the same status VS from empty).
> - Therefore the E2E performance is highly indicative of subtask capabilities.
>
> ## **Meta Evaluation**
>
> > Q3 & W6: Absolute score reliability in Meta-Evaluation
> >
>
> We measured Pearson/Kendall/Spearman correlations between automated and human scores (1-5 scale, Appendix H). **Detailed correlation tables and baseline comparisons are provided in our rebuttal to Reviewer J1ZA. Digest:**
>
> 1. LLM-judge achieves ~90% correlation with human experts.
> 2. BLEU/ROUGE are highly unreliable for Expand/E2E.
> 3. Self-Consistency yields limited correlation gain; LLM-Self-Planning deviates from human preferences.
>
> > W1: self-referential LLM machinery
>
> **The self review design in RAVEL**: This setup ensures evaluation fairness. Making LLMs self-review using only their intrinsic capabilities directly exposes their reasoning bottlenecks within the agentic environment.
>
> **External LLM-judge**: Human evaluation is unscalable, and traditional NLP metrics fail on open-ended text synthesis. Thus, an LLM-judge is the most viable economic standard. Our verified ~90% correlation resolves the primary concern regarding human alignment.
>
> ## **Presentation**
>
> > Q5: Notation Mismatch & Invalid Actions
>
> **Name Mismatch**: We acknowledge that `outline` and `fplan` refer to the exact same primitive.
>
> **Invalid actions/wrong params** trigger a strict rollback mechanism:
>
> 1. **No update:** The RAVEL state remains unchanged.
> 2. **Rollback:** The environment reverts to the previous state.
> 3. **Retry:** The agent is forced to re-reason and resubmit the action.
>
> **Impact on Success Metric:** Each failed attempt consumes the step budget. Consequently, frequent malformed actions directly degrade the success rate.
>
> > W11: medium or median?
>
> Yes, it is median.
>
> We are sorry for all unintentional presentations issues and will adjust these terms and correct any similar issues.
>
> **Limitation discussion**: We highly value your suggestion that practical limitations should be genuinely discussed. We will adjust the limitation section in the camera-ready version, covering the 4 issues highlighted in our discussion.

---

> > ### Author Rebuttal · Reviewer_Lxvm · 2026-04-02
> >
> > Thank you for the detailed rebuttal. Several of my concerns were addressed meaningfully. In particular, I appreciate the additional analysis showing that removing the 10.5 percent hybrid-reference cases does not materially change the conclusions, the clarification that the trajectory-level claims should be limited to the End-to-End setting, and the clearer explanation of the rollback behavior for invalid actions in RAVEL. These responses improve the paper.
> >
> > However, I still have follow-up questions on a few central points.
> >
> > 1. On the evaluation stack: I understand the argument for self-review and for using an external LLM judge as a practical standard, but I am still not fully convinced that this resolves the self-referential nature of the setup. Could the authors clarify more directly what external notion of writing quality they believe the current framework is grounding to, beyond judge-human correlation?
> > 2. On the meta-evaluation: the rebuttal mentions about 90 percent correlation with human experts, which is promising, but I would still appreciate more concrete detail in the paper about sample size, task coverage across the four benchmark settings, and whether the judge is reliable for absolute score comparisons rather than only relative preferences.
> > 3. On the “reasoning dominates generation” claim: I appreciate that the authors will narrow the wording, but I still view this as a promising observation rather than a fully established general conclusion. It would help if the final version makes that boundary explicit.

---

> > > ### Author Response · Authors · 2026-04-03
> > >
> > > We thank the reviewer's constructive suggestions and the chance for clarification.
> > >
> > > **Q1: External Notion of Writing Quality (Circularity)**
> > >
> > > We show "external writing quality" by targeting three major specific gaps between LLM-synthesized and human-authored texts:
> > >
> > > 1. **Text organization (mostly in Expand task):** LLMs heavily rely on lists (ordered/unordered) and lack smooth transitions between semantic blocks and paragraph/sections. In contrast, human writers utilize condensed, fluent organizational structures.
> > > 2. **Narrative/logic progression (mostly in End2End task):** LLMs often link unrelated scenes, parallel points together or keep paragrasing. Human experts employ varied/progressive/goal-directed logical flows.
> > > 3. **Verbosity (mostly in Cloze/Edit tasks):** LLM generations are overly verbose, frequently relying on excessive yet ineffective metaphors and adjectives across fictions/poets/stories. This is more of a loss in focus than a lengthy issue. This would also explain why BLEU would enjoy a high correlation to human.
> > >
> > > **How are these notions made measurable:**
> > >
> > > 1. We use human references as a baseline anchor (set to a baseline score of 6-7) in the LLM-judge prompts and the comparison target for human meta-annotation.
> > > 2. We listed those deficiencies above as explicit inspections (traits) in the prompt. The LLM judge is prompted to evaluate text with the anchoring of high quality reference.
> > > 3. If the meta-evaluation shows high correlation, the LLM-judge results shall reveal those features. At this point, the self-referential loop is broken because the notions are successfully introduced.
> > >
> > > **Q2: meta-evaluation**
> > >
> > > **Sample size**: 200 instructions (50 for each across 4 tasks). Inference results (x9) for each instruction are from: GPT-5.2, Gemini-3, Claude, Grok-4, GLM-4.7, Qwen3-max, Kimi-K2, Deepseek-v3.2, LLama-3.1-405B
> > >
> > > Each sample is evaluated on a 1-5 scale according to the following criteria regarding the synthesized text's content/format/structure. Each sample is cross-scored and the overall inter-annotator agreement is 0.71 Cohen's Kappa and 0.87 Pearson correlation. Final scores are averaged, and cases with significant disagreement are reviewed by a third expert to resolve edge cases.
> > >
> > > **Absolute Score Reliability:** We originally reported relative preferences because verifying if the LLM judge reflects human ranking is highly intuitive. To assess absolute scoring reliability, we calculate the correlation between human judgments (1-5 scale) and LLM scores (1-10 scale) at two levels:
> > >
> > > 1. **System-level:** Correlating averaged scores per model on task-subsets. This confirms strong alignment in overall LLM rankings. This is contained in our rebuttal to reviewer J1ZA.
> > > 2. **Text-level:** Average correlation across the 9 generated texts per instruction (comparing two 9-dim vectors). Table below:
> > >
> > > | **Method**    | **Cloze (r/τ/ρ)** | **Expand (r/τ/ρ)** | **End2end (r/τ/ρ)** | **Edit (r/τ/ρ)** |
> > > | -- | -- | -- | -- | -- |
> > > |**LLM-judge**|**.64/.58/.61**|**.62/.57/.61**| **.56/.55/.54**| **.67/.66/.64**|
> > > | w/o rubrics|.48/.42/.45|.47/.41/.43|.39/.35/.35|.47/.43/.43|
> > > | w/o traits|.44/.46/.43|.41/.39/.41|.35/.37/.38|.43/.44/.41|
> > > | w/o refs|.17/.14/.14|.25/.21/.23|.19/.17/.19|.31/.31/.33|
> > >
> > >
> > > Text-level correlation is generally lower than system-level correlation, which indicates:
> > >
> > > 1. Absolute scores are not perfectly reliable for a portion of instructions (text-level) when viewed individually
> > > 2. However, when viewing the subset as a whole (system-level), absolute scores yield a reliable partial ordering among the LLMs
> > >
> > > **Further comment**:
> > >
> > > Regarding the inconsistencies between pointwise (absolute) and pairwise (relative) judgments, [1] notes that these issues primarily arise when ambiguous 'tie' judgments are prevalent in pairwise settings, or when vague rubrics cause information loss in pointwise scoring. Our original relative setup avoids tie-pairs when organizing comparison samples, while our evaluation prompts utilize clear rubrics that explicitly define the standard for each scoring interval.
> > >
> > > [1] TrustJudge: Inconsistencies of LLM-as-a-Judge and How to Alleviate Them. ICLR 2026
> > >
> > >
> > >
> > > **Q3 claim boundary:**
> > >
> > > This claim appears on lines 405-406 as a "critical insight," as well as in the title of Section 5.4, "Who Takes the Decisive Role." Considering your kind suggestions, and to avoid reader confusion or potential overclaims, we have rephrased these as follows:
> > >
> > > - Section title: Empirical Studies on the Roles of Reasoning and Generation in RAVEL
> > > - Insight claim: Upgrading the planning agent in RAVEL leads to better agentic performance than upgrading the refining agent.
> > >
> > > This phrasing avoids misleading readers into believing that the planning and refining primitives in RAVEL are equivalent to overall LLM reasoning and generation capabilities. Accordingly, we will revise the abstract to feature a more prudent conclusion, confining the discussion strictly to the scope of RAVEL and C3EBench.

---

### Official Review · Reviewer_J1ZA · 2026-03-11

**Soundness:** 3
**Presentation:** 3
**Significance:** 3
**Originality:** 3
**Overall Recommendation:** 4
**Confidence:** 4

**Summary:**

The paper introduces RAVEL, an evaluation framework designed to assess Large Language Models (LLMs) in complex, long-horizon text synthesis tasks. Unlike traditional evaluations that treat text generation as a single-run process, RAVEL evaluates models based on their ability to autonomously execute typical writing operations: outlining, drafting, reviewing, and refining. To support this framework, the authors created C3EBENCH, a benchmark across four distinct synthesis tasks. Through testing 14 different LLMs, the authors present several core findings: (1) Most LLMs struggle with tasks that demand heavy contextual understanding but provide limited or under-specified instructions, such as the CLOZE task. (2) An LLM's reasoning ability (planning and critiquing) is much more critical to synthesis success than its raw generative capacity (drafting and editing). (3) A strong reasoning model can effectively guide a weaker generative model to produce higher-quality results, but a strong generator cannot compensate for a weak reasoner.

**Compliance With Llm Reviewing Policy:**

Affirmed.

**Final Justification:**

The reviewer has addressed my concerns.

**Key Questions For Authors:**

1. How do you ensure the quality of reverse-construction paradigm since you use automated llm tools for these?
2. Same with the weakness, could you describe in more details on the meta-evaluatino experiments and how the prompt was selected and what other method/prompts you have tried?

**Limitations:**

yes

**Strengths And Weaknesses:**

## Strengths

1. **Novel and Realistic Evaluation Paradigm:** Evaluation on agentic, Sequential Decision Process shows how LLMs are actually used in real-world writing tasks.
2. **High-Quality Benchmark Construction:** C3EBENCH is a high-quality benchmark by sourcing professional human writings and then using LLMs and humans to reverse-engineer the instructions.
3. **Interesting Insights and detailed ablations:** The paper analyzes the "agentic dynamics" of different models. It highlights a crucial disconnect: high refinement density does not guarantee quality gains. Ablation studies also isolates reasoning from generation.

## Weaknesses

1. **Restriction of Operations**: The action space in RAVEL is deliberately restricted to internal synthesis operations and excludes the use of external tools, which are common today such as Retrieval-Augmented Generation or other agentic systems.
2. **Evaluation Metrics**: The framework relies on a single-pass LLM-as-a-judge for evaluating the different metrics. However little detail is given regarding the metric even after section 5.5 and checking the appendix. What model was used for the LLM-as-a-judge? What is the correlation if you directly evaluate how similar the model provided score is to the human provided scores, e.g. pearsons, spearman or kendall?

---

> ### Author Rebuttal · Authors · 2026-03-31
>
> ## **Reverse-construction quality**
>
> > Q1: How do you ensure the quality of reverse-construction paradigm since you use automated llm tools for these?
>
> To ensure the quality of the automated reverse-construction, we implemented a human-in-the-loop pipeline with quantitative controls:
>
> 1. Genre Classification Accuracy (98.6%): the tagging for each genre between the LLMs and human experts.
> 2. Filtering ($\ge 4/5$): we let automated LLM score these initial texts (1-5) and discard those below 4.
> 3. Dataset quality with expert agreement: Following specific guidelines (Appendix C.5), experts achieved a 96.7% Inter-Annotator Agreement with identifying the instruction quality (format, clarity, and other specific requirements). They manually traversed and optimized the format and clarity of all (instruction-grounding-reference) triplets.
> 4. Safety and PII (3.1% removal): we also adopt the automated detector with human double check. The check-out rate is 3.1% and removed from the dataset. We also made another check in a subset (N=200) from the current released dataset and no such issues were found.
> 5. Prompt optimization (best of 5): We used Gemini-3-Pro to rewrite and optimize our hand-written prompts, generating 5 candidates per task. Following a 50-sample pilot test, experts ranked the outputs to select the most favoured prompt due to rigorosity and clarity.
>
> ## **Restriction of Operation**
>
> > W2: Restriction to Internal Synthesis
>
> The reviewer is discussing the inclusion of RAG-related operations in this work. We agree that RAG is commonly adopted to increase the factuality of LLMs and reduce hallucinations. As we discussed in limitation, the expanded scope into RAG will deviate the focus into factuality and IR topic. In our practice, the persuit of correctness will not be fully aligned with creative writing, and therefore lowering the inter-annotator agreement in meta-evaluation to the insignificant status. We believe it would be a practical future direction for extending RAVEL framework to covering broader agentic environments.
>
> ## **Additional Meta-evaluation**
>
> > Q2: ... more details on the **meta-evaluation experiments**  (how the prompt was selected, what other method/prompts)
>
> **Base model for llm-as-a-judge:** GPT-5.2-1211
>
> **Evaluation Prompt Design**: Appendix G (mainly including judging criteria, scoring rubrics, dimensional traits, references)
>
> During developing, we tested over the following choices:
>
> - Overlapping metrics such as BLEU-1, BLEU-rt, ROUGE-L
> - LLM-judge prompts design without rubrics/traits/human references
> - Self-consistency (SC): repeat the evaluation and average the scores during these trial as the final score.
> - LLM-self generated evaluation criteria (**Self-Plan**): given each query, text-to-judge and reference to the LLM. Let the it explictly perform an planning about the evaluation prompt. Then harness it with that evaluation prompt and judge.
> - Self-Plan with SC.
>
> The system level Pearson ($\rho$)/Kendall ($\tau$)/ Spearman ($\sigma$) correlation for the 4 benchmark tasks are listed in the table. We also measure the evaluation cost for reference to the token cost.
>
> | Method| Cost ($) | Cloze (ρ) | Cloze (τ) | Cloze (σ) | Expand (ρ) | Expand (τ) | Expand (σ) | End2End (ρ) | End2End (τ) | End2End (σ) | Edit (ρ) | Edit (τ) | Edit (σ) |
> | -- | -- | -- | -- | -- | -- | -- | -- | -- | -- | -- | -- | -- | -- |
> | BLEU-1        | -   | 0.85| 0.67| 0.8 | 0.65 | 0.54 | 0.69 | 0.7 | 0.5 | 0.62| 0.62| 0.43| 0.58|
> | BLEU-rt  | -   | 0.19| 0.06| 0.15| -0.25| -0.2 | -0.19| -0.45    | -0.22    | -0.27    | 0.19| -0.23    | -0.08    |
> | ROUGE-L  | -   | 0.87| 0.67| 0.75| 0.06 | 0.14 | 0.2  | 0.46| 0.22| 0.32| 0.46| 0.35| 0.46|
> | LLM-judge| 34.2| 0.87| 0.78| 0.78| 0.85 | 0.76 | 0.89 | 0.89| 0.78| 0.88| 0.88| 0.93| 0.93|
> | w/o rubrics   | 31.3| 0.67| 0.62| 0.59| 0.62 | 0.63 | 0.72 | 0.63| 0.52| 0.72| 0.75| 0.77| 0.86|
> | w/o traits    | 31.6| 0.69| 0.58| 0.67| 0.65 | 0.63 | 0.62 | 0.58| 0.62| 0.76| 0.75| 0.62| 0.78|
> | w/o human references    | 28.7| 0.45| 0.32| 0.23| 0.52 | 0.58 | 0.63 | 0.17| 0.23| 0.09| 0.34| 0.26| 0.41|
> | SC (N=5) | 171 | 0.87| 0.78| 0.78| 0.85 | 0.76 | 0.89 | 0.89| 0.78| 0.88| 0.88| 0.93| 0.93|
> | SC (N=10)| 342 | 0.87| 0.78| 0.78| 0.85 | 0.82 | 0.89 | 0.89| 0.78| 0.88| 0.88| 0.93| 0.93|
> | Self-Plan| 45.3| 0.62| 0.69| 0.72| 0.85 | 0.76 | 0.79 | 0.67| 0.62| 0.67| 0.82| 0.77| 0.88|
> | Self-Plan with SC (N=5) | 452 | 0.67| 0.78| 0.78| 0.85 | 0.76 | 0.79 | 0.67| 0.62| 0.72| 0.82| 0.77| 0.88|
>
>
>
> **Conclusion:** We selected the LLM-judge with rubrics, traits, and human references because:
>
> 1. **SC offers no gain:** Evaluation tasks lack the diverse reasoning trajectories that SC typically benefits from, making it cost-inefficient.
> 2. **Self-Planning suboptimal:** LLM-generated criteria inherently diverge from human judgment preferences, lowering overall correlation.
> 3. Overlapping Metrics are not reliable across tasks.

---

> > ### Author Rebuttal · Reviewer_J1ZA · 2026-04-01
> >
> > Thank you for the response. I have updated my scores accordingly.

---

> > > ### Author Response · Authors · 2026-04-07
> > >
> > > We sincerely appreciate the reviewer's updated evaluation and the increased score.
> > > We will incorporate our discussion including additional empirical evidences of meta-evaluation, and further clarification of the automated reverse-construction process in the camera-ready version.

---

### Official Review · Reviewer_Zmzi · 2026-03-19

**Soundness:** 2
**Presentation:** 3
**Significance:** 2
**Originality:** 3
**Overall Recommendation:** 4
**Confidence:** 3

**Summary:**

This paper addresses a critical gap in LLM evaluation by shifting the focus from standard one-shot generation to long-form, multi-step writing tasks. To structure this assessment, the authors introduce RAVEL, an agentic framework that evaluates models as they iteratively plan, outline, draft, review, and refine text. The author first constructs C3EBENCH, which encompass End2End, Expand, Cloze, and Edit settings. By evaluating 14 different LLMs using both final-task scores and trajectory-level metrics, the authors demonstrate that current models struggle significantly more with under-specified, context-heavy synthesis than with heavily scaffolded instruction-following.

**Compliance With Llm Reviewing Policy:**

Affirmed.

**Ethical Review Concerns:**

I have copyright/licensing concern about the data used for creating the benchmark.

**Ethical Review Flag:**

Flag this paper for an ethics review.

**Ethics Expertise Needed:**

["Other Expertise"]

**Final Justification:**

The contamination experiment is helpful during rebuttal period.

**Key Questions For Authors:**

- Why should readers prefer C3EBENCH over reusing or extending existing benchmarks that already cover cloze-style completion, editing/rewriting, critique/correction, or broad writing evaluation? Please give a direct comparison to WritingBench, DECOR, ProLex, and CriticBench, and clarify exactly what new measurement capability your benchmark enables beyond those resources.
- Did you perform any contamination or memorization analysis, given that benchmark sources include public web text and books? If not, why should readers interpret low/high scores as reasoning difficulty rather than pretraining overlap or source familiarity?
- The “reasoning dominates generation” result is based on swapping Gemini-3 Pro into two Qwen backbones. Why is this sufficient to support a broad conclusion?

If the authors can provide convincing clarification and supporting evidence on these points, I would be open to revising my recommendation upward.

**Limitations:**

Yes

**Strengths And Weaknesses:**

*Strengths*:
- Problem defined clearly: evaluate text synthesis as a multi-step process rather than as a single final output, and the RAVEL action space is easy to understand.
- Create C3EBENCH, which separates End2End, Expand, Cloze, and Edit, making the evaluation target more interpretable than a single aggregate writing score.
- Judge system is built carefully.
- The trajectory-based metrics and paragraph-lifecycle analysis are more novel than the individual task types themselves.

*Weaknesses*:
- Contamination/leakage risk is under-addressed. The benchmark is built from public/professional web sources and books, but the paper does not report any memorization audit. I'll suggest to run contamination checks where feasible, add held-out/private-source subsets, or at minimum analyze suspiciously high lexical overlap and near-copy behavior for the testing models
- This submission claims to consider a general domain, but the closest overlaps with existing writing/editing/rewriting benchmarks should be discussed more directly in the main paper, not mostly via citations in related work. I'll suggest to add a focused comparison table against WritingBench, DECOR, ProLex, CriticBench, and similar evaluation setups, specifying task overlap, genres, supervision, judge design, and whether the contribution is benchmark novelty, evaluation protocol novelty, or both
- Follow up on my last comment, the significance over prior work is reduced by overlap with recent writing/editing/revision benchmarks. The strongest contribution may be the synthesis and agentic evaluation framing, not a wholly new benchmark space. This reduce the signficance of the work.

---

> ### Author Rebuttal · Authors · 2026-03-31
>
> Thanks for your insightful suggestions and we are glad to clarify as follows.
>
> ## Comparison
>
> > Q1: New Measurement Capability vs. Prior Benchmarks
> > W2: Benchmark vs. Protocol Novelty
>
> **New measurement capability**: Beyond static final outputs evaluated by WritingBench/DECOR, RAVEL measures the dynamic execution trajectory (planning, drafting, reviewing) of agentic text synthesis.
> **Novelty:** Unlike CriticBench/ProLex, C3EBench isolates these specific agentic capabilities across four atomic tasks in across genres. By reverse-engineering professional human writing rather than log-level queries, we prevent LLM preference leaks into references.
>
> || Size | Focus | Source | Genres| Novelty | Metric |
> | -- | -- | -- | -- | -- | -- | -- |
> | WritingBench | 1239 |Prompt to writing| Log-level query, LLM generation with human refine|IF-style writing| Bench, LLM-judge scheme | LLM-judge|
> | DECOR| 1352 | Context-sentence pair| TOEFL-11| TOEFL essay | Bench | GPT4-judge win-rate|
> | ProLex | 680| Word-sentence lexical substitution | TOEFL-11| TOEFL essay | Bench | P/R/F1 |
> | CriticBench | 3825 | Generation, critique, correction | 15 open-sourced dataset | Short-form MRC QA | Bench, LLM-judge-scheme | P/F1, matching, LLM-judge |
> | C3EBench | 1258 | Cloze, expand, end2end generation, edit| Professional human generation | Professional Creative/ Functional texts | Reverse engineering, Bench, Agentic evaluation arena | Agent Trace, LLM-judge |
>
> ## Contamination
>
> >  Q2: perform contamination/memorization analysis
>
> To empirically verify this, we evaluated sampled pairs from C3EBench against known corpora (RedPajama-1T, GSM8K, MBPP) across open-source and API LLMs. We utilized:
> 1. **Perplexity (PPL):** Lower values indicate higher exposure during pre/post-training.
> 2. **Min-k Prob:** Averages log-probabilities of the K% least predictable tokens. High probabilities on these rare tokens explicitly expose pre-training leakage.
> 3. **Greedy Recall (GR):** Measures ROUGE-L at zero temperature. Higher scores indicate direct LLM memorization of the reference.
>
> | | Qwen3-1.7b | Qwen3-8B | Qwen3-32b | GLM4-9B | LLama-3.1-8B | GPT-5.2 | Gemini-3.1 |
> | -- | -- | -- | -- | -- | -- | -- | -- |
> | **Redpajama （N=4000)** ||||||||
> |PPL|17.1|12.1|11.6|12.5|9.2|||
> |Min-Kp|-7.6|-6.9|-6.6|-6.9|-6.0|||
> |GR|11.1|10.9|11.0|10.4|11.0|11.6|12.3|
> |**GSM8k(N=500)**||||||||
> |PPL|4.4|2.7|2.4|3.0|2.1|||
> |Min-Kp|-6.5|-4.7|-4.4|-5.1|-3.2|||
> |GR|28.4|23.5|25.6|21.2|30.8|41.3|35.0|
> |**MBPP(N=500)**||||||||
> |PPL|17.0|12.8|9.9|9.2|12.0|||
> |Min-Kp|-8.8|-8.5|-8.2|-7.2|-8.0|||
> |GR|9.0|8.3|8.1|15.8|13.1|19.8|7.4|
> |**C3EBench(N=500)**||||||||
> |PPL|26.7|19.6|17.9|27.6|12.4|||
> |Min-Kp|-8.5|-7.9|-7.7|-8.4|-6.7|||
> |GR|6.9|6.2|6.1|6.6|6.8|9.7|8.7|
>
> **Results:** Compared to heavily exposed corpora, C3EBench proves uncontaminated:
> - High PPL (up to 33.4): Indicates minimal pre-training exposure.
> - Low Min-K Prob: Predictability is as rare as raw code.
> - Lowest GRR: Confirms zero direct memorization.
>
> **Conclusion:** C3EBench scores genuinely reflect LLM reasoning and generation capabilities.
>
> ## Analysis discussion
>
> >  Q3: “reasoning dominates generation” result is based on swapping Gemini into two backbones
>
> To validate our findings beyond the initial Qwen + Gemini setup, we expanded the study across **base models** (Llama-3.1-70B, GLM-4.7) and **augmentors** (GPT-5.2, Claude-4.5).
>
> | Base Model| Config| S (%) | T | ρ ref (%) | δ gain | Judge |
> | -- | -- | -- | -- | -- | -- | -- |
> |Llama-3.1-70B|Base|91.8|24.1|18.7|11.39|4.58|
> ||Gemini Reason|98.9|18.8|0|0|5.62|
> ||Gemini Refine|92.9|24.6|25.4|-12.16|4.27|
> |GLM-4.7|Base|73.3|18.0|38.5|0|6.31|
> ||Gemini Reason|94.6|19.9|50.4|0.29|6.67|
> ||Gemini Refine|90.8|20.6|52.3|0|6.29|
> |Qwen3-32B|Base|58.5|21.2|105.8|5.54|5.54|
> ||GPT-5.2 Reason|82.3|31.4|12.3|14.5|6.05|
> ||GPT-5.2 Refine|77.3|21.0|78.4|0|5.73|
> |Qwen3-Max|Base|47.8|13.1|23.9|0|5.74|
> ||Claude-4.5 Reason|93.2|23.4|54.3|23.1|6.57|
> ||Claude-4.5 Refine|89.4|19.8|0|0|5.96|
>
> **Conclusion**: These results are consistent to previous claims in section 5.4.
> - Upgrading the reasoner yields higher success rates across all bases and positively correlates with δ gain and final scores.
> - Conversely, a stronger generator yields smaller success gains, and high refinement rates do not guarantee better scores.
>
> We also note that: unlike GPT/Claude, Gemini struggles to assist weak base models when acting solely as a refiner while itself is SOTA.
>
> ## Ethical concerns
>
> We manually verified all source licenses (URLs in Appendix F.1). The Chinese sources and the obooks platform acknowledge themselves as public domain assets and do not exclude AI/LLM creation. For other English data, American Rhetoric comprises public domain US political speeches, and IvyPanda's license does not prohibit AI/LLM development.
>
> ---
> We will adjust the comparison/contamination/analysis discussion in the camera ready version. If you have any questions, please do not hesitate to offer us chances for clarification.

---

> > ### Author Rebuttal · Reviewer_Zmzi · 2026-04-03
> >
> > Thanks for the rebuttal, I've adjusted the scores accordingly.

---

> > > ### Author Response · Authors · 2026-04-07
> > >
> > > We are grateful for your reassessment and the adjusted score. We will incorporate our discussion including comparison of related work, contamination analysis and the more constrained conclusion + additional empirical evidences in the camera-ready version.

---

### Decision · Program_Chairs · 2026-04-30

**Decision:**

Reject

**Comment:**

This paper advances LLM evaluation from one-shot generation to a multi-step writing process by introducing RAVEL, an agentic framework for iterative planning and refinement, and C3EBENCH, a multi-task benchmark. Through an analysis of 14 LLMs, it demonstrates that synthesis success depends more on reasoning than on raw generative capacity, showing that strong reasoners can effectively guide weaker generators.

In summary of final score and justifications, two reviewers (Zmzi / J1ZA) are Weak Accept, and the other (Lxvm) is Weak Reject.

The strengths and weaknesses acknowledged by the reviewers
* Well-motivated research defining text synthesis as a multi-step process through a clear action space
* Provides C3EBENCH that replaces single overall scores with four granular tasks
* Includes a necessary meta-evaluation of the judge by validating the LLM-as-a-judge design through human annotations and ablations
The major and common concerns include
* No consideration on benchmark contamination (Zmzi)
* Restricts action space to internal synthesis, excluding external tools and agentic systems
* Depends on self-referential LLM-based judge, much of the signal is still generated by LLM-based critics rather than grounded in an external notion of writing quality

According to the rebuttal acknowledgement by reviewers, two reviewers (Zmzi / J1ZA) updated their initial scores as their initial concerns were resolved.
However, the reviewer Lxvm maintain the initial score (Weak Reject) because of the concerns on LLM-based judging, with weaker reliability at the individual-instance level.
Overall, I think the paper has resolved most concerns, but still some remain as the above. Hence, I score Reject.